# COMMUNICATION-EFFICIENT AND PRIVATE FEDERATED LEARNING VIA PROJECTED DIRECTIONAL DERIVATIVE

## ABSTRACT

This paper introduces `FedMPDD` (**Fed**erated Learning via **M**ulti-**P**rojected **D**irectional **D**erivatives), a novel algorithm that simultaneously optimizes bandwidth utilization and enhances privacy in Federated Learning. The core idea of `FedMPDD` is to encode each client's high-dimensional gradient by computing its directional derivatives along multiple random vectors. This compresses the gradient into a much smaller message, significantly reducing uplink communication costs from $\mathcal{O}(d)$ to $\mathcal{O}(m)$, where $m \ll d$. The server then decodes the aggregated information by projecting it back onto the same random vectors. Our key insight is that averaging multiple projections overcomes the dimension-dependent convergence limitations of a single projection. We provide a rigorous theoretical analysis, establishing that `FedMPDD` converges at a rate of $\mathcal{O}(1/K)$, matching the performance of FedSGD. Furthermore, we demonstrate that our method provides inherent privacy against gradient inversion attacks due to the geometric properties of low-rank projections, offering a tunable privacy-utility trade-off controlled by the number of projections. Extensive experiments on benchmark datasets validate our theory, showing that `FedMPDD` drastically reduces network congestion and provides strong privacy protection, all while maintaining high model performance, outperforming existing methods in resource-constrained scenarios.

## 1 INTRODUCTION

Federated Learning (FL) is a foundational paradigm for collaboratively training models across $N$ edge devices by leveraging their local computational resources (McMahan et al., 2017; Kairouz et al., 2021; Chen & Ran, 2019) to solve the distributed optimization problem:

$$\text{minimize}_{\mathbf{x} \in \mathbb{R}^d} \ f(\mathbf{x}) = \frac{1}{N} \sum_{i=1}^{N} f_i(\mathbf{x}). \tag{1}$$

Here, $f_i : \mathbb{R}^d \to \mathbb{R}$ is the local objective (loss) function of client $i$, and $f : \mathbb{R}^d \to \mathbb{R}$ is the global objective. Lacking central access to local objectives, FL iteratively communicates between a server and a client subset $\mathcal{A}_k$ ($| \mathcal{A}_k | = \beta N$, where $\beta \in (0, 1]$ denotes the client participation rate). In each round $k$, the server sends the global model $\mathbf{x}_k$. Selected clients compute local updates (e.g., mini-batch gradients $\mathbf{g}_i(\mathbf{x}_k)$) and transmit them to the server. The server aggregates these gradients by averaging: $\mathbf{g}(\mathbf{x}_k) = \frac{1}{\beta N} \sum_{i \in \mathcal{A}_k} \mathbf{g}_i(\mathbf{x}_k)$, and updates the global model: $\mathbf{x}_{k+1} = \mathbf{x}_k - \eta \, \mathbf{g}(\mathbf{x}_k)$, as in `FedSGD` (McMahan et al., 2017), where $\eta$ is a suitable learning rate.

**Privacy Preservation Measures.** Although FL avoids direct data sharing by performing local training, recent studies have shown that a client's raw data can be reconstructed from its transmitted gradients (Zhu et al., 2019; Zhao et al., 2020a; Huang et al., 2021; Yin et al., 2021a; Melis et al., 2019; Li et al., 2022) via a mechanism known as Gradient Inversion Attacks (GIAs). In GIAs within FL, a honest-but-curious server (or any adversary with the same level of server's global information) employs a deep neural network as an inversion model. This inversion network is trained to find input data that, when used to compute gradients, closely matches the gradients shared by legitimate clients, thereby revealing private information (Yin et al., 2021b; Geiping et al., 2020; Jere et al., 2020; Nasr et al., 2019). In FL, a common privacy preservation measure is via a local differential privacy (LDP) framework (Wei et al., 2020; Truex et al., 2020; Zhao et al., 2020b; Seif et al., 2020), where each client adds noise to its gradient before uploading it to the server. By injecting noise locally,

LDP enhances privacy protection without relying on a trusted aggregator. However, this approach introduces a fundamental trade-off between privacy and convergence (Jere et al., 2020).

**Communication Cost Management.** A key bottleneck in FL algorithms like `FedSGD` is the substantial uplink communication overhead from transmitting $d$-dimensional gradients $\mathbf{g}_i(\mathbf{x}_k)$ from clients to the server, requiring $32d$ bits per client per round, assuming single-precision floating-point representation (32 bits per value). For instance, a ResNet-18 model ($\sim 11 \times 10^6$ parameters) necessitates approximately $42$MB transmission per client per round. This high cost severely impacts efficiency in bandwidth-constrained real-world deployments (Chen et al., 2021; Niknam et al., 2020; Shahid et al., 2021; Li et al., 2020). Strategies to reduce FL communication volume or frequency fall into three main classes: *model compression*, *local computation with client selection*, and *gradient compression*. Model compression reduces global model size, for example, by using smaller models with local representations (Liang et al., 2020). Local computation and client selection decrease communication frequency or the number of clients per round through techniques like multiple local updates (Stich, 2018; McMahan et al., 2017; Karimireddy et al., 2020) and client subset selection (Sattler et al., 2019; Liang et al., 2020). Gradient compression reduces the size of transmitted gradients using methods like quantization (Alistarh et al., 2017; Horvóth et al., 2022; Karimireddy et al., 2019; Shlezinger et al., 2020; Bernstein et al., 2018; Reisizadeh et al., 2020; Suresh et al., 2017), sparsification (Ivkin et al., 2019; Lin et al., 2017), and structured/sketched updates (Konečný et al., 2016; Wang et al., 2020; Azam et al., 2021; Han et al., 2024; Ivkin et al., 2019; Cho et al., 2024; Kuo et al., 2024; Qi et al., 2024; Yi et al., 2023; Lin et al., 2022).

**Statement of Contribution.** This paper proposes a novel framework that jointly tackles critical communication efficiency and privacy leakage concerns to enable more practical and widespread adoption of FL. Unlike existing structured projections or sketched updates that primarily focus on compression, our approach introduces a fundamentally new multiplicative encoding paradigm through the *projected directional derivative* within the FL framework. Our algorithm follows the core structure of `FedSGD` but employs the *projected directional derivative* $\hat{\mathbf{g}}_i(\mathbf{x}_k) = \mathbf{u}_{k,i}^\top \mathbf{g}_i(\mathbf{x}_k) \mathbf{u}_{k,i}$ (defined in (1)) instead of the stochastic gradient $\mathbf{g}_i(\mathbf{x}_k)$ to achieve communication efficiency and privacy preservation for each client $i$ through an inherent geometric mechanism that exploits the nullspace properties of low-rank projections. We decompose the projected directional gradient into a scalar directional derivative $\mathbf{u}_{k,i}^\top \mathbf{g}_i(\mathbf{x}_k)$ computed locally by client $i$. Each client independently samples $\mathbf{u}_{k,i}$, then transmits only two scalars: the directional derivative $\mathbf{u}_{k,i}^\top \mathbf{g}_i(\mathbf{x}_k)$ and the random seed $r_{k,i}$ used to generate $\mathbf{u}_{k,i}$. On the server side, the received seed $r_{k,i}$ is used to reconstruct the identical $d$-dimensional vector $\mathbf{u}_{k,i}$, enabling the server to form the gradient estimator $\hat{\mathbf{g}}_i(\mathbf{x}_k)$ without ever transmitting the full vector.

The privacy protection emerges from a novel theoretical insight: the rank deficiency of $\mathbf{u}_{k,i}\mathbf{u}_{k,i}^\top$ in $\hat{\mathbf{g}}_i(\mathbf{x}_k) = \mathbf{u}_{k,i}^\top \mathbf{g}_i(\mathbf{x}_k)\mathbf{u}_{k,i} = \mathbf{u}_{k,i}\mathbf{u}_{k,i}^\top \mathbf{g}_i(\mathbf{x}_k)$ creates an underdetermined system that prevents unique gradient recovery, fundamentally different from differential privacy approaches. The noisy gradient estimator from a single projected directional derivative can degrade performance. To address this, we propose our main algorithm `FedMPDD` (**Fed**erated Learning via **M**ulti-**P**rojected **D**irectional **D**erivatives), which introduces a principled multi-projection aggregation mechanism that averages $m$ projected directional derivatives to estimate the gradient and form our FL algorithm, leading to the following key contributions:

- **Jointly Improved Communication Efficiency and Privacy:** `FedMPDD` significantly reduces uplink communication to $m + 1$ scalars per client per round ($O(m)$ bits, where $m \ll d$). The privacy against GIAs is inherent in the multi-projection encoding due to the rank-deficiency of matrix $\frac{1}{m} U_{k,i}U_{k,i}^\top$, where $U_{k,i}$ is the aggregate matrix of $\{\mathbf{u}_{k,i}^{(j)}\}_{j=1}^m$, with $m$ serving as a tunable parameter for the privacy-communication-accuracy trade-off. Unlike additive noise methods, `FedMPDD` offers a uniform privacy protection regardless of the magnitude of the clients' gradients, eliminating the fluctuating nature of LDP (Remark 3).

- **Enhanced Convergence Rate**: Unlike single-projection approach, `FedMPDD` achieves a convergence rate of $O(1/\sqrt{K})$ (Theorem 2), comparable to standard baselines, through our novel multi-projection averaging mechanism that mitigates the dimensional dependence while preserving the inherent privacy guarantees.

Extensive comparative theoretical analysis and empirical evaluations demonstrate `FedMPDD`'s superior balance of communication efficiency, privacy preservation, and model performance.

**Related Work.** The *projected directional derivative* has been explored in optimization, including balancing computational cost and memory in deep learning gradient calculations (Fournier et al., 2023; Ren et al., 2022; Silver et al., 2021; Baydin et al., 2022) and in zeroth-order optimization (Nesterov & Spokoiny, 2017). To our knowledge, this is the first work to introduce the *projected directional derivative* in FL, specifically through a novel multi-projection decomposition that addresses the dimension-dependent convergence limitations of single-projection methods while achieving joint communication efficiency and privacy preservation. Regarding *joint privacy preservation and communication management*, some communication management techniques offer privacy as a side benefit (Lang et al., 2023; Agarwal et al., 2018). Amiri et al. (Amiri et al., 2021) combined differential privacy (DP) with gradient compression by adding Gaussian noise before quantization, satisfying DP. Lyu et al. (Lyu, 2021) proposed a 1-bit compressor integrating DP into quantization but relies on a fixed quantizer, limiting its adaptability to varying communication budgets. These approaches typically assume a trusted server, unlike LDP, which provides client-side privacy guarantees against potentially curious servers (Chaudhuri et al., 2022).

Our method is fundamentally distinct from structured and sketched updates, which reduce communication by projecting data onto a *fixed*, low-dimensional subspace. Structured updates, such as low-rank adaptation, use a pre-defined subspace for parameters (Hu et al., 2022; Yi et al., 2023; Qi et al., 2024; Zhang et al., 2018; Ullrich et al., 2017; Bertsimas et al., 2023; Cho et al., 2024). Similarly, sketched updates compress gradients using a shared random matrix, fixed at initialization, via techniques like projection matrices (Park & Choi, 2023; Azam et al., 2021; Guo et al., 2024) or Count-Sketch (Ivkin et al., 2019; Rothchild et al., 2020; Haddadpour et al., 2020; Jiang et al., 2018). These approaches rely on a static projection for all clients and rounds, with some variants incurring significant computational overhead to minimize reconstruction error in the fixed subspace (Lin et al., 2022). In contrast, our approach uses a *dynamic* projection strategy based on multi-projected directional derivatives. We compute $\hat{g}_i(x_k) = (u_{k,i}^\top g_i(x_k)) u_{k,i}$, where the projection directions $u_{k,i}$ are randomly and independently sampled for each client $i$ at every round $k$. By averaging multiple projections ($m \ll d$), we overcome the rank-deficiency of a single direction. This mechanism simultaneously achieves significant communication reduction (to $O(m)$ bits per client) and provides inherent privacy through the nullspace effect, representing a fundamental departure from methods reliant on fixed subspaces or post-hoc privacy solutions.

## 2 FL VIA MULTI–PROJECTED DIRECTIONAL DERIVATIVES

The *projected directional derivative* is formally defined as follows.

**Definition 1** (*projected directional derivative*). Let $f : \mathbb{R}^d \to \mathbb{R}$ be a differentiable function. The *projected directional derivative* is defined as $\widehat{\nabla} f(\mathbf{x}) := (\mathbf{u}^\top \nabla f(\mathbf{x})) \mathbf{u}$, where $\nabla f(\mathbf{x})$ is the gradient of $f$ and $\mathbf{u} \in \mathbb{R}^d$ is a random perturbation vector with entries $u_i$ that are independently and identically distributed (i.i.d.) with zero mean and unit variance. □

The projected directional derivative satisfies $\widehat{\nabla} f(\mathbf{x})^\top \nabla f(\mathbf{x}) \geq 0$ and $\mathbb{E}[\widehat{\nabla} f(\mathbf{x})] = \nabla f(\mathbf{x})$ (unbiased estimator for gradient) and thus makes $\mathbf{x}_{k+1} = \mathbf{x}_k - \eta \widehat{\nabla} f(\mathbf{x})$ to behave as a iterative successive descent (i.e., $\mathbb{E}[f(\mathbf{x}_{k+1})|\mathbf{x}_k] \approx f(\mathbf{x}_k) - \eta \|\nabla f(\mathbf{x})\|^2 < f(\mathbf{x}_k)$ for small $\eta > 0$). This stands in contrast to structured/sketched updates, whose gradient estimator $\widetilde{\nabla} f(\mathbf{x})$ is often biased ($\mathbb{E}[\widetilde{\nabla} f(\mathbf{x})] \neq \nabla f(\mathbf{x})$) and can violate the descent condition, thus lacking a general guarantee of progress.

In the context of `FedSGD` we consider an implementation that instead of the stochastic gradient $\mathbf{g}(\mathbf{x}_k)$, we employ the *projected directional derivative* $\hat{\mathbf{g}}(\mathbf{x}_k)$ defined as:

$$\hat{\mathbf{g}}(\mathbf{x}_k) = \frac{1}{\beta N}\Big(\sum_{i \in \mathcal{A}_k} \underbrace{\mathbf{u}_{k,i}^\top \mathbf{g}_i(\mathbf{x}_k)}_{s_i^k:\text{client upload}} \quad \underbrace{\mathbf{u}_{k,i}}_{\text{Server-side projection}}\Big) = \frac{1}{\beta N}\sum_{i \in \mathcal{A}_k} s_i^k \mathbf{u}_{k,i}. \tag{2}$$

Considering the decomposition shown in (2), propose **Fed**erated Learning via **P**rojected **D**irectional **D**erivative (`FedPDD`), see Algorithm 1 in Appendix G, that proceeds as follows: each iteration begins with the server broadcasting the current model $\mathbf{x}_k \in \mathbb{R}^d$ to a sampled client set $\mathcal{A}_k$ (line 4). Upon receipt, each client generates a local random vector $\mathbf{u}_{k,i}$ using its private seed $r_{k,i}$ (line 6). The client encodes its local stochastic gradient into a scalar $s_{k,i}$ using the directional derivative along $\mathbf{u}_{k,i}$, and uploads this scalar together with the seed $r_{k,i}$, which is essential for enabling convergence

---

**Algorithm 2** FedMPDD: **Fed**erated Learning via **M**ulti-**P**rojected **D**irectional **D**erivatives

---

1: **Input:** $\mathbf{x}_0 \in \mathbb{R}^d$, learning rate $\eta$, rounds $K$, # random directions $m$, client fraction $\beta \in (0, 1]$
2: **for** $k = 0, 1, \dots, K - 1$ **do**
3:     Server samples client set $\mathcal{A}_k$ with $|\mathcal{A}_k| = \beta N$
4:     Server broadcasts $\mathbf{x}_k$ to all $i \in \mathcal{A}_k$
5:     **for each** client $i \in \mathcal{A}_k$ **in parallel do**
6:         Compute local stochastic gradient $\mathbf{g}_i(\mathbf{x}_k)$
7:         **for** $j = 1, \dots, m$ **do**             ▷ loop over projected directions
8:             Client generates i.i.d. Rademacher vector $\mathbf{u}_{k,i}^{(j)} \in \{-1, +1\}^d$ using seed $r_{k,i}$ and index $j$
9:             **Encode:** $\mathbf{s}_i^k[j] \leftarrow \big(\mathbf{u}_{k,i}^{(j)}\big)^\top \mathbf{g}_i(\mathbf{x}_k)$
10:         **end for**
11:         Upload $\mathbf{s}_i^k \in \mathbb{R}^m$ and $r_{k,i} \in \mathbb{R}$ to the server
12:     **end for**
13:     $\mathbf{\Delta}_{\text{sum}} \leftarrow \mathbf{0}_d$             ▷ reset the estimator
14:     **for each** client $i \in \mathcal{A}_k$ **do**             ▷ on the server side
15:         **for** $j = 1, \dots, m$ **do**             ▷ re-generate the same projected directions
16:             Server generates i.i.d. Rademacher vector $\mathbf{u}_{k,i}^{(j)} \in \{-1, +1\}^d$ using seed $r_{k,i}$ and index $j$
17:             **Decode:** $\mathbf{\Delta}_{\text{sum}} \leftarrow \mathbf{\Delta}_{\text{sum}} + \frac{\mathbf{s}_i^k[j]}{m} \mathbf{u}_{k,i}^{(j)}$
18:         **end for**
19:     **end for**
20:     **Aggregate:** $\hat{\mathbf{g}}(\mathbf{x}_k) = \frac{1}{\beta N} \mathbf{\Delta}_{\text{sum}}$
21:     **Model update:** $\mathbf{x}_{k+1} = \mathbf{x}_k - \eta \hat{\mathbf{g}}(\mathbf{x}_k)$
22: **end for**
23: **Output:** $\mathbf{x}_K$

---

(line 9). The server aggregates and decodes these scalars to update the global model $\mathbf{x}_{k+1}$ (lines 12-17). The design of FedPDD incorporates two key strategic choices. First, it employs a scalar seed $r_{k,i}$ to generate the identical $d$-dimensional vector $\mathbf{u}_{k,i}$ on the server side, thereby eliminating the need to transmit $\mathbf{u}_{k,i}$ directly while still ensuring convergence. Second, although any zero-mean, unit-variance distribution for $\mathbf{u}_{k,i}$ guarantees an unbiased projected directional derivative, our strategic choice of the Rademacher distribution for generating $\mathbf{u}_{k,i}$, as explained in Appendix Lemma 3, yields lower variance compared to the standard normal distribution.

The preliminary FedPDD algorithm offers two appealing properties. First, it provides significant communication reduction, as clients only transmit two scalars instead of a full gradient vector. Second, it ensures intrinsic privacy preservation due to the rank-deficient nature of the single-vector projection

$$\underbrace{\hat{\mathbf{g}}_i(\mathbf{x}_k)}_{\textbf{Known}} = (\mathbf{u}_{k,i}^\top \mathbf{g}_i(\mathbf{x}_k))\mathbf{u}_{k,i} = \underbrace{(\mathbf{u}_{k,i}\mathbf{u}_{k,i}^\top)}_{\textbf{Known, Rank-1 Matrix}} \mathbf{g}_i(\mathbf{x}_k), \tag{3}$$

which prevents unique gradient reconstruction. However, as our detailed analysis in Appendix B demonstrates, these benefits are offset by poor convergence performance. The high variance introduced by the single, rank-1 projection leads to a convergence rate of $O(d/\sqrt{K})$, which scales poorly with the model dimension $d$. Despite the projected directional derivative being an unbiased estimator, the rank-one map $\mathbf{u}_{k,i}\mathbf{u}_{k,i}^\top$ leads to a magnitude scaling of $\sqrt{d}$ compared to the gradient:

$$\mathbb{E}_{\mathbf{u}}\left[\|\hat{\mathbf{g}}_i(\mathbf{x}_k)\|\right] \leq \sqrt{\mathbb{E}_{\mathbf{u}}\left[\|\hat{\mathbf{g}}_i(\mathbf{x}_k)\|^2\right]} = \sqrt{d\|\mathbf{g}_i(\mathbf{x}_k)\|^2} = \sqrt{d}\,\|\mathbf{g}_i(\mathbf{x}_k)\|.$$

This uncontrollable $\sqrt{d}$ scaling factor leads to higher variance and potential overshooting in FedPDD algorithm. The larger variance of the gradient estimator necessitates a smaller step size (e.g., $\eta = O(1/(d\sqrt{K}))$), significantly slowing convergence (see Remark 3). This critical limitation negates the per-round communication savings over the full training process. Therefore, to address this shortcoming, we introduce FedMPDD (Federated learning with Multi-Projected Directional Derivatives), a generalized version of FedPDD algorithm, as presented in Algorithm 2, which is based on a multi-projection approach.

**Multi-projected directional derivatives.** To address the limitation described above, we extend the estimator by sampling multiple directions. Specifically, at iteration $k$, each selected client $i$ draws $m$ i.i.d. Rademacher vectors $\{\mathbf{u}_{k,i}^{(j)}\}_{j=1}^m \subset \{-1, +1\}^d$. To better understand the mechanism, we now stack the sampled vectors as columns to form the matrix $U_{k,i} \in \mathbb{R}^{d \times m}$. Using this construction, the generalized estimator is defined as:

$$\hat{\mathbf{g}}_i(\mathbf{x}_k) := \frac{1}{m}\sum_{j=1}^m \mathbf{u}_{k,i}^{(j)\top}\mathbf{g}_i(\mathbf{x}_k)\mathbf{u}_{k,i}^{(j)} = \frac{1}{m}U_{k,i}\big(U_{k,i}^\top \mathbf{g}_i(\mathbf{x}_k)\big), \quad U_{k,i} = \begin{bmatrix}\mathbf{u}_{k,i}^{(1)} & \mathbf{u}_{k,i}^{(2)} & \cdots & \mathbf{u}_{k,i}^{(m)}\end{bmatrix} \in \mathbb{R}^{d \times m}$$

As $\mathbb{E}[U_{k,i}U_{k,i}^\top] = mI_d$, the estimator remains unbiased: $\mathbb{E}\big[\hat{\mathbf{g}}_i(\mathbf{x}_k)\big] = \frac{1}{m}\mathbb{E}[U_{k,i}U_{k,i}^\top]\,\mathbf{g}_i(\mathbf{x}_k) = \mathbf{g}_i(\mathbf{x}_k)$.

By constructing, the mapping $\frac{1}{m}U_{k,i}U_{k,i}^\top$ satisfies the high-probability operator-norm Johnson–Lindenstrauss (JL) Lemma (Matoušek, 2008) (Lemma 6 in the Appendix). According to the JL Lemma, if the number of sampled random directions satisfies $m = O\left(\frac{\ln(d/\delta)}{\varepsilon^2}\right)$, then with probability at least $1 - \delta$, the following bound holds:

$$\|\frac{1}{m}\,U_{k,i}\big(U_{k,i}^\top\mathbf{g}_i(\mathbf{x}_k)\big)\| \;\leq\; (1+\varepsilon)\,\|\mathbf{g}_i(\mathbf{x}_k)\|. \tag{4}$$

This result implies that the mapping operator $\frac{1}{m}U_{k,i}U_{k,i}^\top$ approximately preserves the norm of the client's gradient with high probability, provided a sufficient number of sampled directions. Moreover, as $m \to \infty$, the mapping approaches the identity operator $\frac{1}{m}U_{k,i}U_{k,i}^\top \to I_d$ in expectation, due to the unbiasedness of the projection. Motivated by this probabilistic guarantee, which grows only logarithmically with the ambient dimension $d$, we design `FedMPDD` algorithm, presented in Algorithm 2, as a generalization of `FedPDD` algorithm. The following result provides a convergence guarantee for `FedMPDD`.

**Theorem 2** (Convergence Bound of `FedMPDD` Algorithm). *Let the step size $\eta = \frac{1}{L\sqrt{K}}$, and suppose that Assumption 1 holds. Let number of random vectors be $m = O\big(\frac{\ln(d/\delta)}{\varepsilon^2}\big)$. Then, `FedMPDD` algorithm converges to a stationary point of problem* (1) *at a rate of $O(1/\sqrt{K})$, satisfying the following upper bound with probability at least $1 - \delta$,*

$$\frac{1}{K}\sum_{k=0}^{K-1}\mathbb{E}\left[\|\nabla f(\mathbf{x}_k)\|^2\right] \leq \underbrace{O\left(\frac{L(f(\mathbf{x}_0)-f^\star)}{K^{0.5}}\right)}_{\text{due to initialization}} + \underbrace{O\left(\frac{\sigma^2(1/\beta - 1)}{K^{1.5}}\right)}_{\text{due to client sampling}} + \underbrace{O\left(\frac{\epsilon G^2}{K^{0.5}}\right)}_{\text{due to Multi-projected directional derivatives}} , \tag{5}$$

*where $0 < \epsilon < 1$ is the distortion parameter, $\beta \in (0, 1]$ denotes the client participation fraction, and $f^\star$ denotes the global minimum of $f$.*

**Remark 1** (Computational Cost of `FedMPDD`). The client-side encoding in `FedMPDD` has a computational cost of $O(dm)$ (see lines 7–10 of the `FedMPDD` algorithm; as reported in Table A.10 for one representative experiment, this computational time is negligible and does not constitute a bottleneck in our experiments). While this may initially seem costly, it is often offset in practice, since in many federated learning settings, client models are deep neural networks and computing the full stochastic gradient (line 6) is already expensive. Recent work (Baydin et al., 2022; Ren et al., 2022; Silver et al., 2021) has shown that computing the inner product $\mathbf{u}^\top\mathbf{g}_i$ is significantly more efficient than computing the full gradient $\mathbf{g}_i$, because the operation can be implemented as a Jacobian-vector product (JVP), which leverages efficient vector-matrix multiplication in deep networks. Specifically, projected-forward methods reduce the time complexity of gradient computation from $O(h^2pT^2)$ (for full forward-mode autodiff) to $O(h^2T + hpT)$, where $h$, $p$, and $T$ denote the hidden dimension, the number of parameters per layer, and the total number of layers, respectively. Motivated by these insights, `FedMPDD` can avoid computing $\mathbf{g}_i$ explicitly (line 6) by fixing a single mini-batch $\mathcal{B}_i^k$ and reusing it across all random directions. The encoding step (line 9) is then performed via the projected-forward approach using JVPs. We can show that when $m < \frac{hpT}{h+p}$, this strategy reduces overall client-side computation, making `FedMPDD` particularly suitable for resource-constrained devices. We empirically evaluate this strategy in our follow-up study (see Section F). For further details on the computational and memory complexity of the projected-forward approach, see Table. F.1. □.

**Communication Reduction and Efficiency in `FedMPDD`:** FedMPDD presented in Algorithm 2 significantly reduces per-round uplink communication by enabling clients to transmit only an $m$-dimensional vector $\mathbf{s}_i^k \in \mathbb{R}^m$ together with a scalar seed number $r_{k,i} \in \mathbb{R}$, instead of the full $d$-dimensional gradient ($m \ll d$). This is achieved by encoding the client's $d$-dimensional gradient through its projection onto a set of $m$ random scalars (line 9). Moreover, the total communication cost over the full training horizon is reduced to $O(1/\sqrt{K} \times \beta N \times m)$, where $\beta$ is the client participation ratio and $N$ is the number of clients. Since $m$ grows only logarithmically with the problem dimension $d$, the communication savings become even more substantial for large-scale models.

**Intrinsic Privacy Preservation**: The privacy guarantees of `FedMPDD` are demonstrated under a standard honest-but-curious threat model, which provides a formal basis for our analysis.

**Definition 2** (*Threat Model*). An **honest-but-curious** adversary (e.g., the server) correctly follows the protocol but attempts to infer private client data by analyzing all accessible information. This includes communication messages, model architecture, and global hyperparameters.

Against this adversary, `FedMPDD`'s privacy stems from its rank-deficient projection ($m \ll d$), which creates a quantifiable uncertainty for any party observing the transmitted data. We formalize this protection below.

**Lemma 1** (Gradient Reconstruction Error). For the reconstructed gradient estimator $\hat{\mathbf{g}}_i(\mathbf{x}_k) = \frac{1}{m} \sum_{j=1}^{m} \mathbf{u}_{k,i}^{(j)} (\mathbf{u}_{k,i}^{(j)})^\top \mathbf{g}_i(\mathbf{x}_k)$, the expected relative squared error between the reconstructed and true stochastic gradients is given by:

$$\mathbb{E}_U \left[ \|\hat{\mathbf{g}}_i(\mathbf{x}_k) - \mathbf{g}_i(\mathbf{x}_k)\|^2 \right] / \|\mathbf{g}_i(\mathbf{x}_k)\|^2 = \frac{d-1}{m}. \tag{6}$$

This inherent gradient ambiguity provides a formal defense against GIAs by establishing a lower bound on an adversary's ability to reconstruct the original private input data.

**Lemma 2** (Lower Bound on Private Data Reconstruction Error). Suppose an adversary attempts to reconstruct a private input vector $v$ by minimizing the loss between the observed projected gradient and a dummy gradient, $\mathcal{L}(\hat{v}) := \| \frac{1}{m} U_{k,i} U_{k,i}^\top \mathbf{g}_i(v, c; \mathbf{x}_k) - \mathbf{g}_i(\hat{v}, c; \mathbf{x}_k) \|$. The expected reconstruction error for the attack-optimal output $\hat{v}^*$ is lower bounded by:

$$\mathbb{E}\left[ \|v - \hat{v}^*\|^2 \right] \geq \frac{d-1}{m \cdot L_v(\mathbf{x})^2} \|\mathbf{g}_i(v, c; \mathbf{x}_k)\|^2, \tag{7}$$

where $L_v(\mathbf{x})$ is the Lipschitz constant of the gradient with respect to $v$.

Together, these lemmas establish a direct link between our projection mechanism and a concrete privacy guarantee. The gradient reconstruction error of $\frac{d-1}{m}$ translates into a formal lower bound on data recovery, creating a privacy barrier that scales with model dimension $d$. Our approach offers fundamental advantages over additive-noise methods like Local Differential Privacy (LDP). In LDP, the privacy level is inconsistent, as its relative reconstruction error is proportional to $1/\|\mathbf{g}_i(\mathbf{x}_k)\|^2$ as shown in Remark 5 in Appendix C. This creates a dilemma: large gradients are poorly protected, while small gradients can be overwhelmed by noise, harming model convergence. Achieving consistent privacy with LDP would require

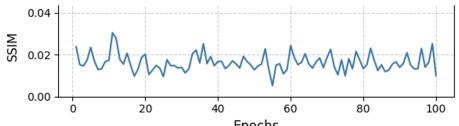

Figure 1: The SSIM scores from the GIA (Yu et al., 2025) on the LeNet model, using the *projected directional derivative* estimator with $m = 600$ in `FedMPDD`, remain consistently low (below 0.04) over 100 training epochs. As showed by Lemma 1 and 2, this demonstrates that the privacy level remains stable and is independent of the training stage.

large, performance-degrading noise values. In contrast, `FedMPDD` provides a consistent relative reconstruction error of $\frac{d-1}{m}$, which is independent of the gradient's magnitude. This design simultaneously ensures: (i) consistent privacy without harming utility (by preserving the descent direction), and (ii) high communication efficiency. For a detailed derivation and discussion, please see Appendix C and E.

Our theoretical guarantees are supported by empirical results. For example, Fig. 2 illustrates the `FedMPDD` Algorithm's data obfuscation on a client's private data for different values of $m$, as detailed in our numerical examples section. Similar to the observation in Remark 5, `FedMPDD` inherently provides a consistent relative reconstruction error of $(d-1)/m$ at each communication round. This consistent privacy benefit, independent of gradient size, can be observed in Fig. 1 in our numerical study (detailed in Appendix A).

**Remark 2** (Multi-Round Privacy Composition). A key consideration is privacy erosion from an adversary observing a client over multiple rounds. Our formal analysis in Appendix D shows that even in a worst-case scenario (e.g., a static gradient), unique gradient recovery is impossible as long as the total number of observed projections is less than the model's dimension. Specifically, privacy is guaranteed if $T \times m < d$, where $T$ is the number of rounds. While the natural evolution of gradients during training provides stronger practical protection, this bound establishes a fundamental privacy guarantee for our method. $\square$

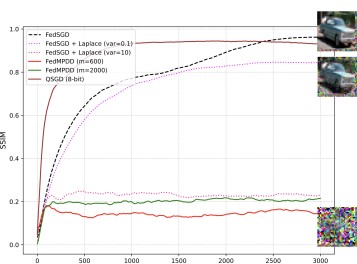 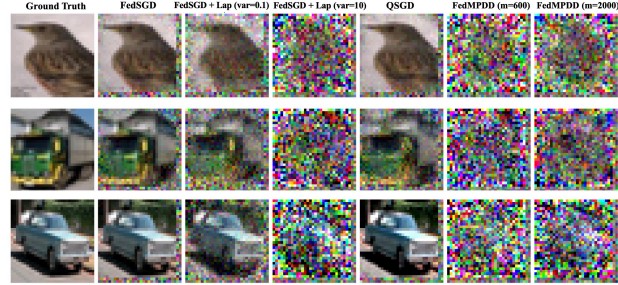

Figure 2: GIA attack (Yu et al., 2025) visualization: SSIM scores (left) and reconstructed CIFAR-10 samples (right) across methods. LDP with small noise (columns 3-4) and QSGD (column 5) show significant data leakage. `FedMPDD` demonstrates stronger privacy (columns 6 and 7).

**Privacy-Communication Trade-off**: The parameter $m$ serves as a tunable knob for the privacy-communication-accuracy trade-off. A larger value of $m$ directly translates to higher communication overhead per round, as the uplink message size scales with $m$. Conversely, as shown in Lemmas 1 and 2, a larger $m$ improves the accuracy of reconstructed gradients, reflected in the reduced expected reconstruction error of $\frac{d-1}{m}$. Regarding privacy, a larger $m$ implies less privacy, as more information about the original gradient is revealed during the projection process. This multi-faceted relationship highlights a fundamental trade-off: improving one aspect (e.g., accuracy) often comes at the expense of another (e.g., privacy or communication efficiency). This mirrors similar trade-offs observed in differential privacy, where stronger privacy guarantees typically lead to a reduction in model utility or require higher communication/computation. The worst-case bound $T < d/m$ directly quantifies this trade-off: smaller $m$ allows for more training rounds while maintaining privacy, but may require more communication rounds to achieve convergence. Our experimental results across two attack families, including the recent gradient inversion attack (Yu et al., 2025) and the well-known deep leakage from gradients method (Zhu et al., 2019), support this theoretical finding and demonstrate that the predicted privacy protection holds in practice.

## 3 NUMERICAL EXPERIMENTS

**Experimental Setup.** We conducted experiments on three standard machine learning and FL benchmark datasets using four model architectures with varying parameter sizes (For details see Appendix H.1). To comprehensively evaluate performance, we tested client participation rates of 10%, 50%, and 100% across different tasks, considering both IID and non-IID data distributions (where each client accesses only a subset of classes in multi-class classification). Hyperparameter tuning details for each model are in Appendix H.2. For gradient inversion attacks, we employed two algorithms: (i) the recent method proposed by Yu et al. (2025), and (ii) the well-known Deep Leakage from Gradients (DLG) algorithm (Zhu et al., 2019), which reconstructs original input data (e.g., images) from shared gradients in distributed learning.

**Performance vs. Joint Communication Efficiency and Privacy Leakage Mitigation.** To highlight the communication cost reduction of `FedMPDD`, we compare it against a recent sketching-based method (Lin et al., 2022), a structured-based method (Yang et al., 2024), a top-$k$ sparsification method (Alistarh et al., 2018), and the quantization-based method QSGD (Alistarh et al., 2017). For performance evaluation, FedSGD serves as the accuracy baseline. Our communication cost analysis includes total and per-round uplink overhead, as well as *i)* performance under a constrained communication budget and *ii)* the total communication cost to achieve target accuracy. To empirically validate `FedMPDD`'s privacy enhancement against GIAs, we compare it to LDP with varying noise levels in image classification tasks.

To evaluate the quality of reconstructed images after the attack, we employ the Structural Similarity Index Measure (SSIM) (Lang et al., 2023), a widely used metric for assessing image similarity, where an SSIM value closer to 1 indicates a higher resemblance between the reconstructed image and the ground truth. Due to space limitations, we only show the subset of the results and the full set of table and training and accuracy curves will be presented in Appendix A. Note that in our experiments, we did not fine-tune the value of $m$ to explicitly optimize for the minimal communication cost and

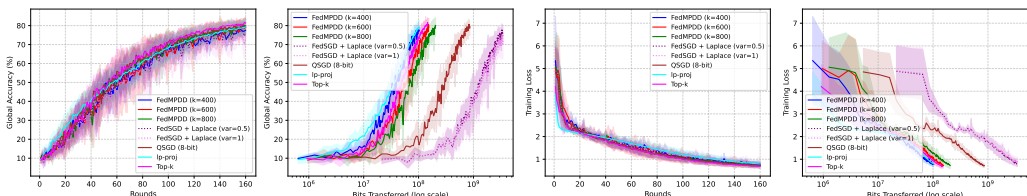

Figure 3: Training loss and accuracy curves versus communication rounds and number of transmitted bits for the LeNet model on the MNIST dataset (IID).

Table 1: Comparison of test accuracy (under a fixed communication budget), communication cost (under a target accuracy), privacy leakage, and reconstruction quality using the attack of (Yu et al., 2025) on MNIST (IID) with LeNet.

| Method | Bytes Budget (GB) | Test Acc (%) | Target Acc (%) | Used Bytes (GB) | Defendability | SSIM |
|---|---|---|---|---|---|---|
| FedSGD | 0.09 | 11.45 | 60 | 1.439 | ✗ | 1.00 |
| FedSGD + Laplace (var=0.5) | 0.09 | 11.13 | 60 | 1.611 | ✓ | ≪ 0.03 |
| FedSGD + Laplace (var=1) | 0.09 | 11.41 | 60 | 1.869 | ✓ | ≪ 0.03 |
| **FedMPDD (m=400, 2% d)** | 0.09 | **77.37** | 60 | **0.052** | ✓ | ≪ 0.03 |
| **FedMPDD (m=600, 3% d)** | 0.09 | **67.75** | 60 | **0.079** | ✓ | ≪ 0.03 |
| **FedMPDD (m=800, 4% d)** | 0.09 | **58.49** | 60 | **0.093** | ✓ | ≪ 0.03 |
| QSGD (8-bit) | 0.09 | 21.66 | 60 | 0.376 | ✗ | 0.98 |
| Top-k (k=400) | 0.09 | 65.75 | 60 | 0.077 | ✗ | 0.89 |
| lp-proj | 0.09 | 73.01 | 60 | 0.069 | ✗ | 0.75 |

maximal privacy guarantees achievable by `FedMPDD`. Instead, we selected $m = O\left(\frac{\ln(d/\delta)}{\varepsilon^2}\right)$ for sufficiently small values of $\delta$ and $\varepsilon$. In Table A.9 of Appendix A, we present experiments over a wide range of $m$ values to further illustrate the findings of Theorem 2. As $m$ becomes too small, both the rate of convergence and the accuracy deteriorate. Moreover, as we show in our reported results, the chosen values of $m$ grow slightly with the parameter dimension $d$ (ranging from a simple logistic model to a deep CNN model with over 300,000 parameters) while maintaining convergence performance comparable to FedSGD, making the proposed algorithm well-suited for large-scale problems. This empirical observation further supports the theoretical guarantees in Theorem 2, where $m$ is required to grow logarithmically with the dimension $d$ to retain the $O(1/\sqrt{K})$ convergence rate of FedSGD.

Tables 1 and 2 demonstrate `FedMPDD`'s effectiveness in jointly reducing communication cost and enhancing privacy. Our byte budget represents the *total uplink communication* permitted between active clients and the server across all training iterations, unlike per-round limits. We analyze the results from two complementary perspectives, beginning with those reported in Table 2.

**Fixed budget (0.9 GB):** In Table 2 FedSGD and its Laplace-noised variants rapidly exceed the communication budget in the very first iteration, making them impractical under realistic constraints. In contrast, `FedMPDD` stays well within the budget, achieving competitive accuracy thanks to its efficient projected directional derivative encoding. For example, with $m = 600$ (0.2% of $d$), `FedMPDD` reaches $40.8\%$ test accuracy, significantly higher than QSGD (12.9%) and other baselines such as lp-proj (34.7%) (Lin et al., 2022), Top-k (38.1%) (Alistarh et al., 2018), and SA-FedLora (35.8%) (Yang et al., 2024). Importantly, although these baselines remain within the budget, they fail to provide consistent privacy guarantees, as their SSIM values (0.74 to 0.91) reveal substantial leakage under gradient inversion attacks. In contrast, `FedMPDD` achieves both stronger accuracy and substantially lower SSIM (0.14 to 0.22), highlighting its ability to simultaneously reduce communication and preserve privacy, rigorously established in Lemmas 1 and 2, arising from the $(d - m)$-dimensional nullspace of the multi-projected directional derivative."

**Fixed accuracy (60% target):** To achieve the same target accuracy, in Table 2 FedSGD and its noisy variants consume over 470 GB, exceeding the budget by several orders of magnitude and leaking private information (SSIM > 0.8). Similarly, QSGD requires more than 117 GB while still failing on privacy. lp-proj, Top-k, and SA-FedLora are more communication-efficient, which is their primary goal, requiring only 1.8 to 2.3 GB, but they still exhibit weak privacy protection due to their high SSIM values. By contrast, `FedMPDD` with $m = 600$ requires only 1.3 GB, representing a **more than**

Table 2: Comparison of test accuracy (under a fixed communication budget), communication cost (under a target accuracy), privacy leakage, and reconstruction quality using the attack of (Yu et al., 2025) on CIFAR10 (IID) with CNN model from (McMahan et al., 2017). $\star$ indicates budget exceeded in the first iteration.

| Method | Bytes Budget (GB) | Test Acc (%) | Target Acc (%) | Used Bytes (GB) | Defendability | SSIM |
|---|---|---|---|---|---|---|
| FedSGD | 0.90 | $\star$ | 60 | 471.96 | ✗ | 0.96 |
| FedSGD + Laplace (var=0.1) | 0.90 | $\star$ | 60 | 471.96 | ✗ | 0.84 |
| FedSGD + Laplace (var=10) | 0.90 | $\star$ | 60 | not reached | ✓ | 0.23 |
| **FedMPDD (m=600, 0.2 % d)** | 0.90 | **40.84** | 60 | **1.32** | ✓ | 0.14 |
| **FedMPDD (m=2000, 0.6 % d)** | 0.90 | **36.26** | 60 | **3.26** | ✓ | 0.22 |
| QSGD (8-bit) | 0.90 | 12.97 | 60 | 117.98 | ✗ | 0.93 |
| lp-proj | 0.90 | 34.72 | 60 | 1.84 | ✗ | 0.74 |
| Top-k (k=600) | 0.90 | 38.11 | 60 | 2.30 | ✗ | 0.91 |
| SA-FedLora | 0.90 | 35.84 | 60 | 2.10 | ✗ | 0.83 |

**356$\times$ reduction** compared to FedSGD, and with $m = 2000$, it requires just 3.3 GB, still a **144$\times$ reduction**. Crucially, FedMPDD attains these communication savings while keeping SSIM $< 0.22$, ensuring strong and constant privacy level.

Taken together, these results demonstrate that FedMPDD outperforms all baselines across both evaluation criteria: it matches or exceeds their communication efficiency while uniquely combining this with robust privacy protection. Competing methods (lp-proj, Top-k, SA-FedLora) achieve communication reduction but fail on privacy, whereas FedMPDD achieves both simultaneously.

Figure 2 illustrates FedMPDD's privacy-preserving strength under the GIA (Yu et al., 2025). The left plot shows SSIM scores over iterations, and the right panel visualizes reconstructed CIFAR-10 samples. Laplace noise with variance 0.1 (a typical LDP setting) fails to protect data, yielding high SSIM and clear reconstructions, while variance 10 provides privacy but severely degrades model accuracy. In contrast, FedMPDD with $m = 2000$ achieves a comparable privacy level to Laplace(10) without adding noise, since its protection arises from the $(d - m)$-dimensional nullspace of the projection, rigorously analyzed in Lemmas 1 and 2. At the same time, it reduces per-round communication by more than **150$\times$**, highlighting FedMPDD's dual benefit of strong privacy and efficiency.

While increasing $m$ accelerates convergence, it also incurs higher communication cost and potentially greater privacy leakage (as expected from Lemma 1 and 2, increasing $m$ decreases the inherent privacy protection at a rate of $O(1/m)$, resulting in higher SSIM scores and more successful image reconstructions). However, as illustrated, for instance, in Fig. A.9 in the appendix, smaller values of $m$ can actually achieve comparable or even faster convergence to the target accuracy, while simultaneously offering stronger privacy guarantees as a beneficial side effect. This makes FedMPDD particularly suitable for large-scale problems where both privacy and communication efficiency are critical. This behavior can be intuitively explained by the nullspace effect of the *projected directional derivative* mechanism, which effectively suppresses certain components of noise in the stochastic gradient, thereby stabilizing the optimization. For additional visualizations of another attack model (Zhu et al., 2019) across different architectures, as well as full training and accuracy curves under various methods, please refer to Appendix A.

## 4 CONCLUSION

We introduced FedMPDD, a novel FL framework addressing communication efficiency and privacy leakage through a gradient encoding and decoding mechanism based on multi-projected directional derivatives. Building upon the single-projection FedPDD, which offered initial communication and privacy benefits but suffered from dimension-dependent convergence, FedMPDD averaged multiple projections to achieve comparable convergence rates to baselines. Our theoretical analysis and empirical evaluations demonstrated FedMPDD's superior balance of communication cost, performance, and privacy, facilitated by its efficient gradient encoding and decoding. We achieved significant uplink communication reductions compared to baseline methods, including structured, sketched, quantized, and sparsified approaches, while simultaneously ensuring robust and uniform privacy against GIAs, unlike the fluctuating and often weak privacy guarantees of LDP. The tunable parameter $m$ allowed flexible trade-offs. Notably, smaller $m$ values sometimes yielded faster convergence with stronger privacy. For future work and further discussions see Appendix F.

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

APPENDIX

## A    ADDITIONAL EXPERIMENTAL RESULTS

This section presents additional experimental results on both IID and non-IID data distributions, evaluating the proposed method in terms of communication cost reduction and privacy preservation across various datasets and models with varying dimensionalities.

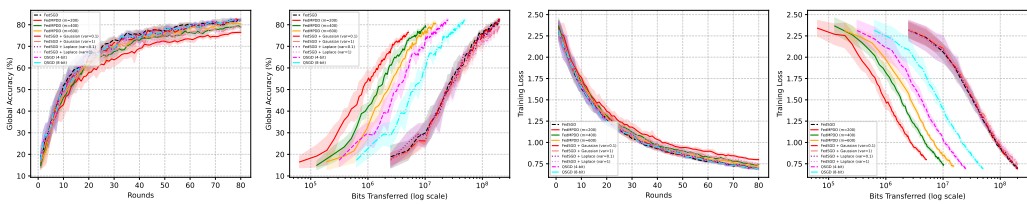

Figure A.1: Training loss and accuracy curves versus communication rounds and number of transmitted bits for the logistic model on the MNIST dataset (IID)..

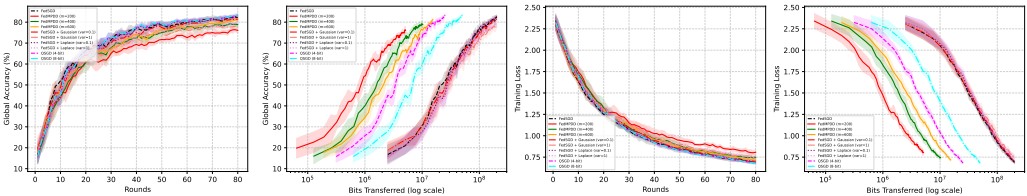

Figure A.2: Training loss and accuracy curves versus communication rounds and number of transmitted bits for the logistic model on the MNIST dataset (non-IID)..

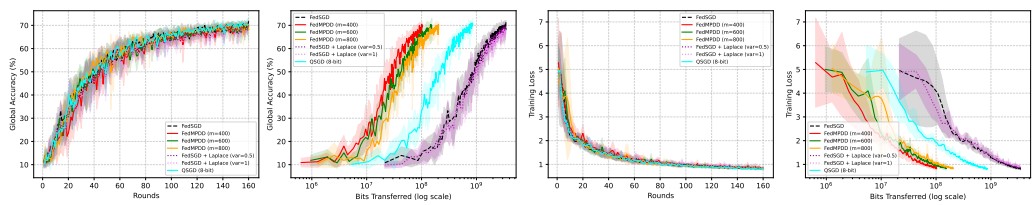

Figure A.3: Training loss and accuracy curves versus communication rounds and number of transmitted bits for the LeNet model on the FMNIST dataset (IID)..

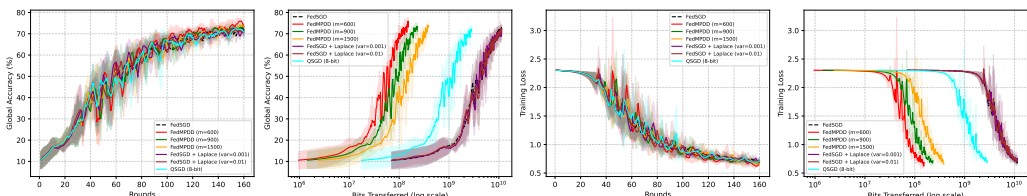

Figure A.4: Training loss and accuracy curves versus communication rounds and number of transmitted bits for the CNN model from (Lin et al., 2022) on the FMNIST dataset (IID).

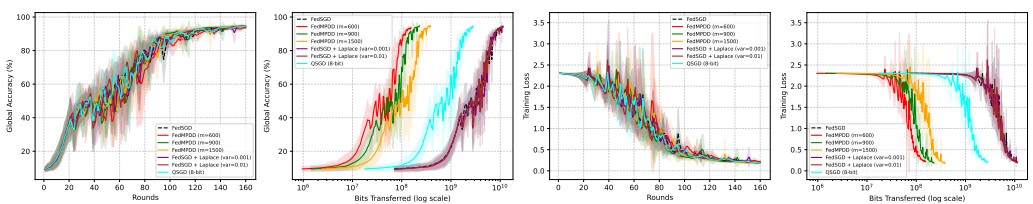

Figure A.5: Training loss and accuracy curves versus communication rounds and number of transmitted bits for the CNN model from (Lin et al., 2022) on the MNIST dataset (IID).

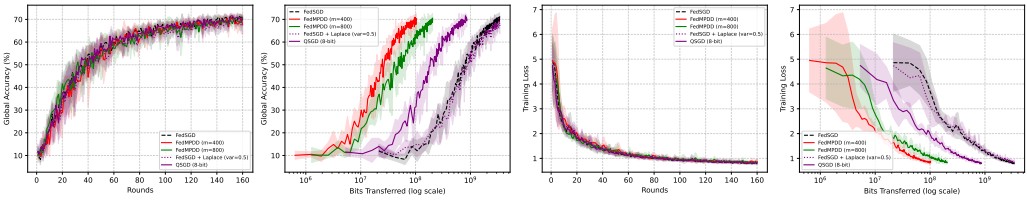

Figure A.6: Training loss and accuracy curves versus communication rounds and number of transmitted bits for the LeNet model on the FMNIST dataset (non-IID).

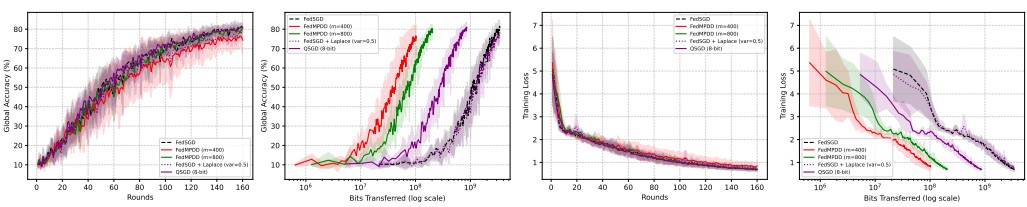

Figure A.7: Training loss and accuracy curves versus communication rounds and number of transmitted bits for the LeNet model on the MNIST dataset (non-IID).

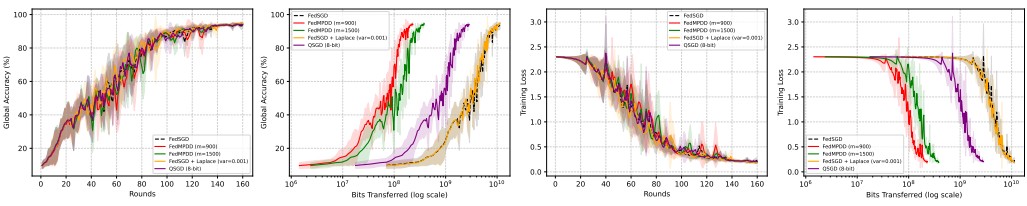

Figure A.8: Training loss and accuracy curves versus communication rounds and number of transmitted bits for the CNN model from (Lin et al., 2022) on the MNIST dataset (non-IID).

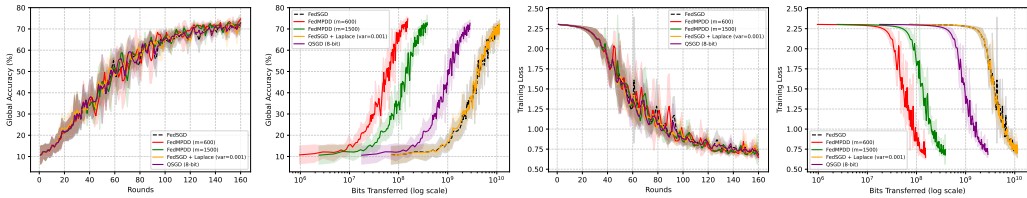

Figure A.9: Training loss and accuracy curves versus communication rounds and number of transmitted bits for the CNN model from (Lin et al., 2022) on the FMNIST dataset (non-IID).

Table A.1: Comparison of test accuracy under a fixed communication budget, communication usage for target accuracy, privacy leakage, and reconstruction quality on MNIST (IID) using the attack of (Zhu et al., 2019) with a logistic model. A ⋆ in the Test Acc column indicates that the communication budget was already exceeded in the first iteration of the algorithm.

| Method | Bytes Budget | Test Acc | Target Acc | Used Bytes | Defendability | SSIM |
|---|---|---|---|---|---|---|
| FedSGD | 2000000 | ⋆ | 0.6 | 40192000.0 | ✗ | 1.00 |
| FedSGD + Gaussian (var=0.1) | 2000000 | ⋆ | 0.6 | 42704000.0 | ✗ | 0.80 |
| FedSGD + Gaussian (var=1) | 2000000 | ⋆ | 0.6 | 45216000.0 | ✗ | 0.59 |
| FedSGD + Laplace (var=0.1) | 2000000 | ⋆ | 0.6 | 42704000.0 | ✗ | 0.82 |
| FedSGD + Laplace (var=1) | 2000000 | ⋆ | 0.6 | 45216000.0 | ✗ | 0.60 |
| **FedMPDD (m=200)** | 2000000 | **65.29** | 0.6 | **1536000.0** | ✓ | 0.03 |
| **FedMPDD (m=400)** | 2000000 | **57.62** | 0.6 | **2432000.0** | ✓ | 0.14 |
| **FedMPDD (m=600)** | 2000000 | **48.85** | 0.6 | **3456000.0** | ✓ | 0.13 |
| QSGD (4-bit) | 2000000 | 38.92 | 0.6 | 5343440.0 | ✗ | 0.88 |
| QSGD (8-bit) | 2000000 | 28.86 | 0.6 | 10681440.0 | ✗ | 0.99 |

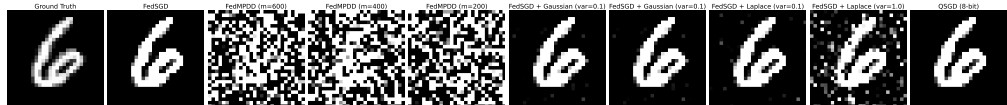

Figure A.10: Attack results on logistic model using MNIST dataset.

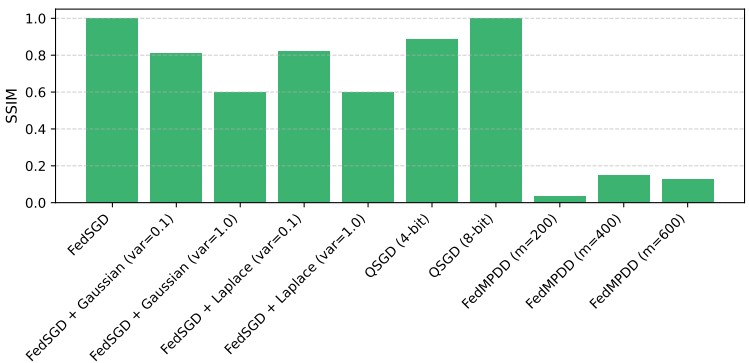

Figure A.11: SSIM bar plot on logistic model using MNIST dataset.

Table A.2: Comparison of test accuracy under a fixed communication budget, communication usage for target accuracy, privacy leakage, and reconstruction quality on FMNIST (IID) using the attack of (Zhu et al., 2019) with LeNet model.

| Method | Bytes Budget | Test Acc | Target Acc | Used Bytes | Defendability | SSIM |
|---|---|---|---|---|---|---|
| FedSGD | 90000000 | 15.85 | 0.6 | 1331859200.0 | ✗ | 1.00 |
| FedSGD + Laplace (var=0.5) | 90000000 | 15.76 | 0.6 | 1245932800.0 | ✓ | ≪ 0.03 |
| FedSGD + Laplace (var=1) | 90000000 | 15.91 | 0.6 | 1525193600.0 | ✓ | ≪ 0.03 |
| **FedMPDD (m=400)** | 90000000 | **66.77** | 0.6 | **44160000.0** | ✓ | ≪ 0.03 |
| **FedMPDD (m=600)** | 90000000 | **66.10** | 0.6 | **61440000.0** | ✓ | ≪ 0.03 |
| **FedMPDD (m=800)** | 90000000 | **64.84** | 0.6 | **78080000.0** | ✓ | ≪ 0.03 |
| QSGD (8-bit) | 90000000 | 25.73 | 0.6 | 311576000.0 | ✗ | 0.99 |

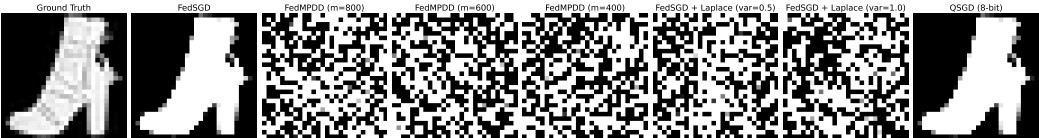

Figure A.12: Attack results on LeNet using FMNIST dataset.

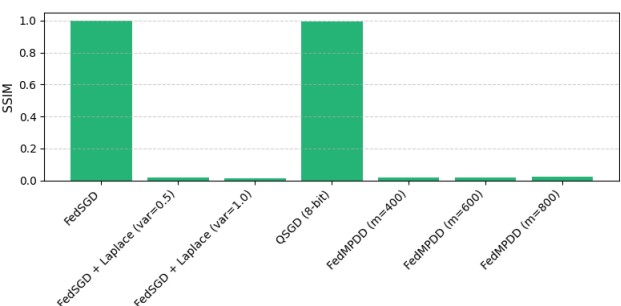

Figure A.13: SSIM bar plot on LeNet using FMNIST dataset.

Table A.3: Comparison of test accuracy under a fixed communication budget, communication usage for target accuracy, and privacy leakage on MNIST (non-IID) using the attack of (Yu et al., 2025) with LeNet model.

| Method | Bytes Budget | Test Acc | Target Acc | Used Bytes | Defendability |
|---|---|---|---|---|---|
| FedSGD | 90000000 | 11.06 | 0.6 | 1460748800.0 | ✗ |
| FedSGD + Laplace (var=0.5) | 90000000 | 12.75 | 0.6 | 1482230400.0 | ✓ |
| **FedMPDD (m=400)** | 90000000 | **76.00** | 0.6 | **51840000.0** | ✓ |
| **FedMPDD (m=800)** | 90000000 | **57.80** | 0.6 | **93440000.0** | ✓ |
| QSGD (8-bit) | 90000000 | 18.66 | 0.6 | 359924000.0 | ✗ |

Table A.4: Comparison of test accuracy under a fixed communication budget, communication usage for target accuracy, and privacy leakage on FMNIST (non-IID) using the attack of (Zhu et al., 2019) with LeNet model.

| Method | Bytes Budget | Test Acc | Target Acc | Used Bytes | Defendability |
|---|---|---|---|---|---|
| FedSGD | 90000000 | 12.05 | 0.6 | 1288896000.0 | ✗ |
| FedSGD + Laplace (var=0.5) | 90000000 | 15.06 | 0.6 | 1224451200.0 | ✓ |
| **FedMPDD (m=400)** | 90000000 | **69.47** | 0.6 | **37760000.0** | ✓ |
| **FedMPDD (m=800)** | 90000000 | **61.18** | 0.6 | **74240000.0** | ✓ |
| QSGD (8-bit) | 90000000 | 24.30 | 0.6 | 306204000.0 | ✗ |

Table A.5: Comparison of test accuracy under a fixed communication budget, communication usage for target accuracy, privacy leakage, and reconstruction quality on MNIST (IID) using the attack of (Yu et al., 2025) with CNN model from (Lin et al., 2022).

| Method | Bytes Budget | Test Acc | Target Acc | Used Bytes | Defendability | SSIM |
|---|---|---|---|---|---|---|
| FedSGD | 150000000 | 10.40 | 0.6 | 3980569600.0 | ✗ | 0.81 |
| FedSGD + Laplace (var=0.001) | 150000000 | 10.30 | 0.6 | 4193814400.0 | ✗ | 0.48 |
| FedSGD + Laplace (var=0.01) | 150000000 | 10.33 | 0.6 | 3625161600.0 | ✓ | 0.13 |
| **FedMPDD (m=600)** | 150000000 | **93.26** | 0.6 | **64320000.0** | ✓ | 0.17 |
| **FedMPDD (m=900)** | 150000000 | **91.46** | 0.6 | **86400000.0** | ✓ | 0.28 |
| **FedMPDD (m=1500)** | 150000000 | **59.46** | 0.6 | **146400000.0** | ✓ | 0.28 |
| QSGD (8-bit) | 150000000 | 14.56 | 0.6 | 977460000.0 | ✗ | 0.64 |

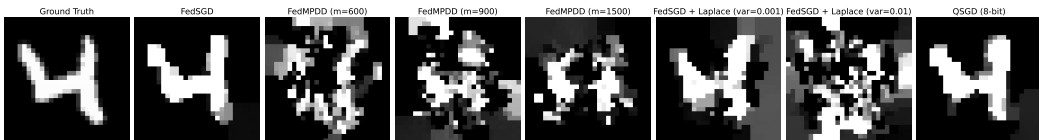

Figure A.14: Attack results on CNN model from (Lin et al., 2022) using MNIST dataset.

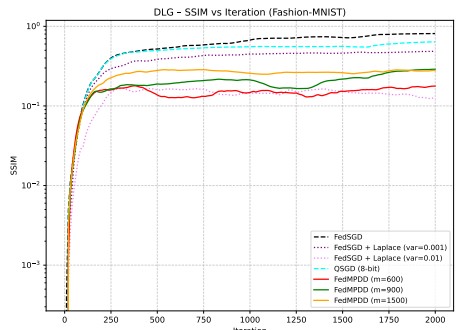

Figure A.15: SSIM plot versus iteration on CNN model from (Lin et al., 2022) using MNIST dataset.

Table A.6: Comparison of test accuracy under a fixed communication budget, communication usage for target accuracy, privacy leakage, and reconstruction quality on FMNIST (IID) using the attack of (Yu et al., 2025) with CNN model from (Lin et al., 2022).

| Method | Bytes Budget | Test Acc | Target Acc | Used Bytes | Defendability |
|---|---|---|---|---|---|
| FedSGD | 150000000 | 11.74 | 0.6 | 5757609600.0 | ✗ |
| FedSGD + Laplace (var=0.001) | 150000000 | 11.95 | 0.6 | 4975712000.0 | ✓ |
| FedSGD + Laplace (var=0.01) | 150000000 | 12.48 | 0.6 | 5402201600.0 | ✓ |
| **FedMPDD (m=600)** | 150000000 | **75.81** | 0.6 | **73920000.0** | ✓ |
| **FedMPDD (m=900)** | 150000000 | **62.36** | 0.6 | **103680000.0** | ✓ |
| **FedMPDD (m=1500)** | 150000000 | **47.32** | 0.6 | **199200000.0** | ✓ |
| QSGD (8-bit) | 150000000 | 15.51 | 0.6 | 1421760000.0 | ✗ |

Table A.7: Comparison of test accuracy under a fixed communication budget, communication usage for target accuracy, and privacy leakage on MNIST (non-IID) using the attack of (Zhu et al., 2019) with CNN model from (Lin et al., 2022).

| Method | Bytes Budget | Test Acc | Target Acc | Used Bytes | Defendability |
|---|---|---|---|---|---|
| FedSGD | 200000000 | 11.40 | 0.6 | 3696243200.0 | ✗ |
| FedSGD + Laplace (var=0.001) | 200000000 | 11.04 | 0.6 | 3696243200.0 | ✗ |
| **FedMPDD (m=900)** | 200000000 | **93.08** | 0.6 | **90720000.0** | ✓ |
| **FedMPDD (m=1500)** | 200000000 | **80.67** | 0.6 | **151200000.0** | ✓ |
| QSGD (8-bit) | 200000000 | 21.84 | 0.6 | 1030776000.0 | ✗ |

Table A.8: Comparison of test accuracy under a fixed communication budget, communication usage for target accuracy, privacy leakage, and reconstruction quality on FMNIST (non-IID) using the attack of (Zhu et al., 2019) with CNN model from (Lin et al., 2022).

| Method | Bytes Budget | Test Acc | Target Acc | Used Bytes | Defendability |
|---|---|---|---|---|---|
| FedSGD | 150000000 | 12.22 | 0.6 | 5828691200.0 | ✗ |
| FedSGD + Laplace (var=0.001) | 150000000 | 12.03 | 0.6 | 5615446400.0 | ✓ |
| **FedMPDD (m=600)** | 150000000 | **70.83** | 0.6 | **74880000.0** | ✓ |
| **FedMPDD (m=1500)** | 150000000 | **51.32** | 0.6 | **199200000.0** | ✓ |
| QSGD (8-bit) | 150000000 | 13.51 | 0.6 | 1457304000.0 | ✗ |

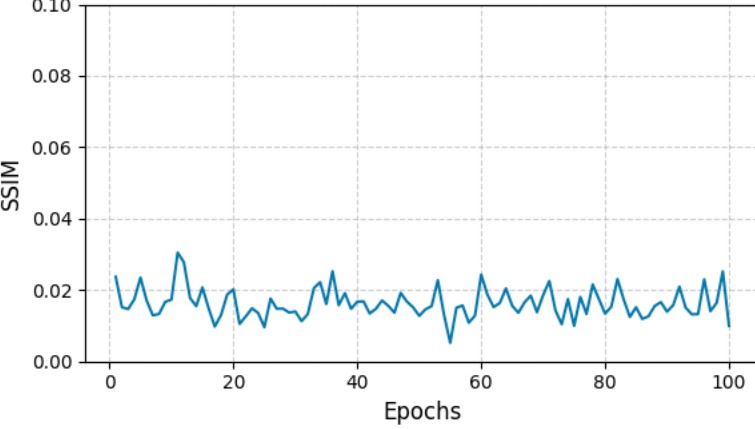

Figure A.16: The SSIM scores produced by the attack in (Yu et al., 2025) on the LeNet model using the *projected directional derivative* estimator $\hat{\mathbf{g}}_i(\mathbf{x})$ with $m = 600$ in `FedMPDD` remain uniformly low over 100 training epochs, confirming that `FedMPDD` enforces a constant privacy guarantee throughout the entire training process.

Table A.9: `FedMPDD` performance with varying numbers of random directions $m$ on the MNIST dataset using the LeNet model.

| Method | Test Acc. (%) |
|---|---|
| **FedMPDD** (m=50) | 30.44 |
| **FedMPDD** (m=200) | 75.02 |
| **FedMPDD** (m=400) | 77.52 |
| **FedMPDD** (m=600) | 79.02 |

Table A.10: Per-round, per-client latency (ms) to compute $\mathbf{s}_i^k[j] = (\mathbf{u}_{k,i}^{(j)})^\top \mathbf{g}_i(\mathbf{x}_k)$, averaged over clients and rounds on LeNet–MNIST.

| # random directions $m$ | Avg. latency (ms) |
|---|---|
| 400 | 0.43 |
| 600 | 0.61 |
| 800 | 0.93 |

# B    DETAILS ON FEDPDD: COMMUNICATION, PRIVACY ANALYSIS AND CON-VERGENCE RATE

In this section, we provide additional details on the design, communication cost, and intrinsic privacy-preserving properties of the FedPDD algorithm. We first show how uplink communication is drastically reduced compared to FedSGD, then explain the inherent privacy guarantee derived from the rank-deficient structure of the projected directional derivative. Finally, we present the convergence analysis of FedPDD, including the role of projection in the convergence bound, along with its implications for large-scale settings.

**Uplink Communication Reduction:** FedPDD achieves significant per-iteration uplink communication reduction over FedSGD. Achieved by strategically decomposing the directional derivative, FedPDD enables each client to transmit only two scalars per iteration: (i) the directional derivative $s_i^k$ (32 bits), instead of a full $d$-dimensional gradient vector ($32d$ bits), and (ii) the seed $r_{k,i}$ (32 bits), which allows the server to regenerate $\mathbf{u}_{k,i}$ consistently. This design decouples the uplink communication cost from the problem dimension $d$, thereby alleviating bandwidth constraints in high-dimensional or resource-limited settings.

**Intrinsic privacy preservation attribute of `FedPDD`**: The intrinsic privacy preservation of FedPDD stems from the challenge of recovering $\mathbf{g}_i(\mathbf{x}_k)$ from the server's received information. This requires solving the following linear algebraic equation:

$$\underbrace{\hat{\mathbf{g}}_i(\mathbf{x}_k)}_{\text{Known}} = (\mathbf{u}_{k,i}^\top \mathbf{g}_i(\mathbf{x}_k))\mathbf{u}_{k,i} = \underbrace{(\mathbf{u}_{k,i}\mathbf{u}_{k,i}^\top)}_{\text{Known, Rank-1 Matrix}} \mathbf{g}_i(\mathbf{x}_k). \tag{8}$$

As $\mathbf{u}_{k,i}\mathbf{u}_{k,i}^\top$ is a rank-1 matrix whose nullspace is of dimension $d-1$ (Sylvester's theorem (Horn & Johnson, 2012)), (8) does not have a unique solution. Any $\mathbf{g}_i(\mathbf{x}_k) + \mathbf{v}$ for $\mathbf{v} \in \ker(\mathbf{u}_{k,i}\mathbf{u}_{k,i}^\top)$ also satisfies the equation. This $d-1$ dimensional nullspace (which scales linearly with parameter dimension $d$) inherently introduces a high degree of uncertainty, making it impossible to uniquely reconstruct the true local gradient. A more detailed discussion regarding privacy preservation will be provided in section 2. This is because, while FedPDD offers inherent privacy, its reliance on a single projected directional derivative leads to a suboptimal performance, motivating the need for an enhanced approach.

To analyze the convergence behavior of FedPDD, we first introduce some widely used conventional assumptions (T Dinh et al., 2020; Liu et al., 2022) on the cost function, similar to those used in the analysis of FedSGD.

**Assumption 1** (Regularity Conditions on Local Objectives). For each client $i$, the local function $f_i$ is $L$-smooth, i.e., for all $\mathbf{x}, \mathbf{y} \in \mathbb{R}^d$, $\|\nabla f_i(\mathbf{x}) - \nabla f_i(\mathbf{y})\| \le L \|\mathbf{x} - \mathbf{y}\|$. In addition, the stochastic gradient is bounded in second moment: $\mathbb{E}\big[\|\mathbf{g}_i(\mathbf{x})\|^2\big] \le G^2$ for all $\mathbf{x} \in \mathbb{R}^d$, and the variance of local gradients relative to the global gradient is bounded: $\frac{1}{N}\sum_{i=1}^N \|\nabla f_i(\mathbf{x}) - \nabla f(\mathbf{x})\|^2 \le \sigma^2$.  $\square$

**Theorem 1** (Convergence Bound of `FedPDD` Algorithm). *Consider FedPDD algorithm with a step size $\eta = \frac{1}{L\sqrt{K}}$, and suppose that Assumption 1 holds. Then, FedPDD algorithm converges to a stationary point of problem* (1) *at a rate of $O(d/\sqrt{K})$, satisfying the following upper bound*

$$\frac{1}{K}\sum_{k=0}^{K-1} \mathbb{E}\left[\|\nabla f(\mathbf{x}_k)\|^2\right] \le \underbrace{O\left(\frac{L(f(\mathbf{x}_0) - f^\star)}{K^{0.5}}\right)}_{\text{due to initialization}} + \underbrace{O\left(\frac{\sigma^2(1/\beta - 1)}{K^{1.5}}\right)}_{\text{due to client sampling}} + \underbrace{O\left(\frac{dG^2}{K^{0.5}}\right)}_{\text{due to projected directional derivative}}, \tag{9}$$

*where $\beta \in (0, 1]$ denotes the client participation fraction, and $f^\star$ denotes the global minimum of $f$.*

**Remark 3** (Implications of Stepsize Choice on the Convergence Bound (9)). The third term in the convergence bound (9) stems from the rank-1 projection operator $\mathbf{u}_{k,i}\mathbf{u}_{k,i}^\top$, as shown in the proof of Theorem 1 in Appendix E. While adjusting the step size to $\eta = \frac{1}{dL\sqrt{K}}$ could mitigate the $d$ factor in the projection-induced term of the convergence bound (9), it would necessitate a smaller overall step size, potentially slowing convergence as indicated by the increased influence of the other terms. This trade-off motivates the development of a dimension-independent extension in section 2 for practical large-scale problems.  $\square$

**Remark 4** (Communication Cost Analysis of `FedPDD`). Despite `FedPDD`'s per-round uplink efficiency in reducing bandwidth usage, its overall communication cost over training is comparable to FedSGD. The $O(d/\sqrt{K})$ convergence rate of `FedPDD`, versus $O(1/\sqrt{K})$ for FedSGD, implies roughly $d$ times more iterations for similar performance. Consequently, the total uplink communication volume for `FedPDD` becomes $O(d/\sqrt{K} \times \beta N)$ bits, matching FedSGD's asymptotic cost.

## C  PRIVACY PRESERVATION: FEDMPDD VS. LOCAL DIFFERENTIAL PRIVACY

**Remark 5** (Privacy Preservation Attributes of the `FedMPDD` Algorithm vs. LDP). In FL with LDP, adding noise $\zeta_i$ to each client's gradient introduces a relative reconstruction error at the server, quantified as

$$
\text{Relative Reconstruction Error} \propto \frac{\mathbb{E}_{\zeta_i}\left[\|\overbrace{\mathbf{g}_i(\mathbf{x}_k) + \zeta_i}^{\substack{\text{honest-but-curious} \\ \text{server observation}}} - \mathbf{g}_i(\mathbf{x}_k)\|^2\right]}{\|\mathbf{g}_i(\mathbf{x}_k)\|^2} = \frac{\mathbb{E}_{\zeta_i}\left[\|\zeta_i\|^2\right]}{\|\mathbf{g}_i(\mathbf{x}_k)\|^2} = \frac{d\,\tau^2}{\|\mathbf{g}_i(\mathbf{x}_k)\|^2}, \tag{10}
$$

where $\tau^2$ denotes the variance of the added Gaussian or Laplace noise. This means that if a client's gradient $\mathbf{g}_i(\mathbf{x}_k)$ is small, the added noise dominates and privacy appears stronger. Conversely, large gradients are less effectively masked by the same noise level. Achieving consistent privacy across varying gradient magnitudes would require significantly increasing $\tau^2$, but this directly harms learning and final accuracy. In more details, additive-noise LDP schemes face three key limitations: (1) they may break convergence when $\|\zeta\| \gg \|\mathbf{g}\|$, since the perturbed gradient $\bar{\mathbf{g}} = \mathbf{g} + \zeta$ can flip the descent direction ($\mathbf{g}^\top \bar{\mathbf{g}} < 0$), or lose privacy when $\|\zeta\| \ll \|\mathbf{g}\|$; (2) their privacy depends on $\|\mathbf{g}\|$, hiding $\mathbf{g}$ only when the noise is sufficiently large; and (3) they require sending the full $d$-dimensional noisy gradient, incurring $O(d)$ uplink communication cost.

In contrast, our `FedMPDD` algorithm, as shown in Lemma 1 and 2, inherently ensures a consistent relative reconstruction error of $\frac{d-1}{m}$ per round. This provides (i) unbiasedness and guaranteed descent ($\mathbf{g}^\top \hat{\mathbf{g}} \geq 0$), (ii) constant privacy through a fixed $(d-m)$-dimensional null space independent of $\|\mathbf{g}\|$, and (iii) substantial communication savings by transmitting only an $m \ll d$ dimensional projection. Unlike additive-noise mechanisms, this guarantee scales with model complexity ($d$) and is intrinsic to the projected directional derivative design, without requiring explicit noise that degrades performance.

$\square$

## D    MULTI-ROUND PRIVACY COMPOSITION

The critical challenge arises when an adversary aggregates observations across multiple rounds. Let $\mathcal{S}_i^T = \{\mathbf{s}_i^1, \mathbf{s}_i^2, \ldots, \mathbf{s}_i^T\}$ denote the sequence of projections observed from client $i$ over $T$ rounds.

**Theorem 2** (Worst-Case Multi-Round Privacy Bound). *Consider the worst-case scenario where the client's gradient $g_i$ remains constant across all rounds. An adversary observing projections from $T$ rounds obtains $T \times m$ linear constraints on the $d$-dimensional gradient. Privacy is preserved (gradient cannot be uniquely recovered) as long as:*

$$T < \frac{d}{m}$$

*due to the underdetermined system having a non-trivial nullspace of dimension $d - T \times m > 0$.*

*Proof.* In the worst case where $g_i(x^k) = g$ is constant, each round $k$ provides $m$ linear equations of the form $\langle g, u_i^{(j)} \rangle = s_i^{(j)}$ for $j = 1, \ldots, m$. After $T$ rounds, the adversary has $T \times m$ linear constraints on the $d$-dimensional vector $g$. The system is underdetermined when $T \times m < d$, ensuring that multiple solutions exist in the $(d - T \times m)$-dimensional nullspace, preventing unique gradient recovery. $\qquad\square$

**Remark 6** (Practical Implications). *This worst-case analysis provides a conservative bound for privacy protection. In practice, gradients evolve during training, potentially providing additional privacy protection through the changing gradient manifold. However, the bound $T < d/m$ establishes a fundamental limit for privacy preservation under the most adversarial conditions.*

# E  DETAILS AND PROOFS

Below we present omitted proofs and lemmas for the main results presented in the paper.

**Lemma 1** (Gradient Reconstruction Error). For the reconstructed gradient estimator $\hat{\mathbf{g}}_i(\mathbf{x}_k) = \frac{1}{m} \sum_{j=1}^{m} \mathbf{u}_{k,i}^{(j)}(\mathbf{u}_{k,i}^{(j)})^\top \mathbf{g}_i(\mathbf{x}_k)$, the expected relative squared error between the reconstructed and true stochastic gradients is given by:

$$\mathbb{E}_U \left[ \|\hat{\mathbf{g}}_i(\mathbf{x}_k) - \mathbf{g}_i(\mathbf{x}_k)\|^2 \right] / \|\mathbf{g}_i(\mathbf{x}_k)\|^2 = \frac{d-1}{m}. \tag{11}$$

*Proof.* Note that for a single direction $\mathbf{u}_{k,i}$ ($m = 1$),

$$\left\|\hat{\mathbf{g}}_i(\mathbf{x}_k)\right\|^2 = \mathbf{u}_{k,i}^\top(\mathbf{u}_{k,i}^\top \mathbf{g}_i(\mathbf{x}_k))(\mathbf{u}_{k,i}^\top \mathbf{g}_i(\mathbf{x}_k))\mathbf{u}_{k,i} = (\mathbf{u}_{k,i}^\top \mathbf{u}_{k,i})(\mathbf{g}_i(\mathbf{x}_k)^\top \mathbf{u}_{k,i}\mathbf{u}_{k,i}^\top \mathbf{g}_i(\mathbf{x}_k)).$$

Since for Rademacher $\mathbf{u}_{k,i}$, we have $\mathbf{u}_{k,i}^\top \mathbf{u}_{k,i} = d$, it follows that

$$\mathbb{E}_{\mathbf{u}}\left[\|\hat{\mathbf{g}}_i(\mathbf{x}_k)\|^2\right] = d\,\mathbf{g}_i(\mathbf{x}_k)^\top \mathbb{E}_{\mathbf{u}}[\mathbf{u}_{k,i}\mathbf{u}_{k,i}^\top]\,\mathbf{g}_i(\mathbf{x}_k) = d\,\|\mathbf{g}_i(\mathbf{x}_k)\|^2.$$

Therefore, since the estimator satisfies

$$\mathbb{E}_{\mathbf{u}}[\hat{\mathbf{g}}_i(\mathbf{x}_k)] = \mathbf{g}_i(\mathbf{x}_k),$$

we obtain for one direction

$$\mathbb{E}_{\mathbf{u}}\left[\|\hat{\mathbf{g}}_i(\mathbf{x}_k) - \mathbf{g}_i(\mathbf{x}_k)\|^2\right] = \mathbb{E}_{\mathbf{u}}\left[\|\hat{\mathbf{g}}_i(\mathbf{x}_k)\|^2 - \|\mathbf{g}_i(\mathbf{x}_k)\|^2\right] = (d-1)\|\mathbf{g}_i(\mathbf{x}_k)\|^2.$$

Now extend to $m$ directions. The multi-direction estimator is

$$\hat{\mathbf{g}}_i(\mathbf{x}_k) = \frac{1}{m} \sum_{j=1}^{m} \mathbf{u}_{k,i}^{(j)}(\mathbf{u}_{k,i}^{(j)})^\top \mathbf{g}_i(\mathbf{x}_k).$$

Because the $u_{k,i}^{(j)}$ are independent and identically distributed, the cross terms vanish in expectation, and all diagonal terms are the same. Thus,

$$\mathbb{E}\left[\|\hat{\mathbf{g}}_i(\mathbf{x}_k) - \mathbf{g}_i(\mathbf{x}_k)\|^2\right] = \frac{1}{m}(d-1)\|\mathbf{g}_i(\mathbf{x}_k)\|^2.$$

Dividing both sides by $\|\mathbf{g}_i(\mathbf{x}_k)\|^2$ yields in (11). $\qquad\square$

**Lemma 2** (Lower Bound on Private Data Reconstruction Error). Suppose an adversary attempts to reconstruct a private input vector $v$ by minimizing the loss between the observed projected gradient and a dummy gradient, $\mathcal{L}(\hat{v}) := \|\frac{1}{m}U_{k,i}U_{k,i}^\top \mathbf{g}_i(v, c; \mathbf{x}_k) - \mathbf{g}_i(\hat{v}, c; \mathbf{x}_k)\|$. The expected reconstruction error for the attack-optimal output $\hat{v}^*$ is lower bounded by:

$$\mathbb{E}\left[\|v - \hat{v}^*\|^2\right] \geq \frac{d-1}{m \cdot L_v(\mathbf{x})^2}\|\mathbf{g}_i(v, c; \mathbf{x}_k)\|^2, \tag{12}$$

where $L_v(\mathbf{x})$ is the Lipschitz constant of the gradient with respect to $v$.

*Proof.* Consider the reconstruction loss

$$\mathcal{L}(\hat{v}) = \left\|\frac{1}{m}U_{k,i}U_{k,i}^\top \mathbf{g}_i(v, c; \mathbf{x}_k) - \mathbf{g}_i(\hat{v}, c; \mathbf{x}_k)\right\|$$

$$= \left\|\left(\mathbf{g}_i(v, c; \mathbf{x}_k) - \mathbf{g}_i(\hat{v}, c; \mathbf{x}_k)\right) - \left(\mathbf{g}_i(v, c; \mathbf{x}_k) - \frac{1}{m}U_{k,i}U_{k,i}^\top \mathbf{g}_i(v, c; \mathbf{x}_k)\right)\right\|$$

$$\geq \left| \|\mathbf{g}_i(v, c; \mathbf{x}_k) - \frac{1}{m}U_{k,i}U_{k,i}^\top \mathbf{g}_i(v, c; \mathbf{x}_k)\| - \|\mathbf{g}_i(v, c; \mathbf{x}_k) - \mathbf{g}_i(\hat{v}, c; \mathbf{x}_k)\| \right|$$

$$\geq \|(I - \frac{1}{m}U_{k,i}U_{k,i}^\top)\mathbf{g}_i(v, c; \mathbf{x}_k)\| - \|\mathbf{g}_i(v, c; \mathbf{x}_k) - \mathbf{g}_i(\hat{v}, c; \mathbf{x}_k)\|.$$

Thus,

$$\|\mathbf{g}_i(v, c; \mathbf{x}_k) - \mathbf{g}_i(\hat{v}, c; \mathbf{x}_k)\| \geq \mathrm{proj}_g(v, c; \mathbf{x}_k, U_{k,i}) - \mathcal{L}(\hat{v}),$$

where
$$\text{proj}_g(v, c; \mathbf{x}_k, U_{k,i}) := \|(I - \tfrac{1}{m} U_{k,i} U_{k,i}^\top)\, \mathbf{g}_i(v, c; \mathbf{x}_k)\|$$
denotes the projection-induced gradient error.

Assuming that for fixed $\mathbf{x}_k$ and known $c$, the map $v \mapsto \mathbf{g}_i(v, c; \mathbf{x}_k)$ is $L_v(\mathbf{x}_k)$-Lipschitz (as is satisfied by standard models), we obtain
$$\|v - \hat{v}\| \ \geq\ \frac{\text{proj}_g(v, c; \mathbf{x}_k, U_{k,i}) - \mathcal{L}(\hat{v})}{L_v(\mathbf{x}_k)}.$$

With the convention that the right-hand side is interpreted as $\max\{0, \cdot\}$, in particular for an attack-optimal recovery $\hat{v}^\star \in \arg\min_{\hat{v}} \mathcal{L}(\hat{v})$,
$$\|v - \hat{v}^\star\| \ \geq\ \frac{\text{proj}_g(v, c; \mathbf{x}_k, U_{k,i}) - \mathcal{L}(\hat{v}^\star)}{L_v(\mathbf{x}_k)}, \qquad \text{and if } \mathcal{L}(\hat{v}^\star) = 0, \ \ \|v - \hat{v}^\star\| \ \geq\ \frac{\text{proj}_g(v, c; \mathbf{x}_k, U_{k,i})}{L_v(\mathbf{x}_k)}.$$

Squaring the above bound, taking expectation with respect to $U_{k,i}$, and invoking Lemma 1, we arrive at
$$\mathbb{E}[\|v - \hat{v}^\star\|^2] \ \geq\ \frac{1}{L_v(\mathbf{x}_k)^2}\, \mathbb{E}\big[\text{proj}_g(v, c; \mathbf{x}_k, U_{k,i})^2\big] \ =\ \frac{d - 1}{m\, L_v(\mathbf{x}_k)^2}\, \|\mathbf{g}_i(v, c; \mathbf{x}_k)\|^2.$$

Here, we used the fact that from Lemma 1, i.e.,
$$\mathbb{E}\big[\text{proj}_g(v, c; \mathbf{x}_k, U_{k,i})^2\big] = \frac{d - 1}{m}\, \|\mathbf{g}_i(v, c; \mathbf{x}_k)\|^2.$$

$\square$

**Lemma 3** (Variance Reduction via Distribution Choice for $\mathbf{u}$ in (2).). *Consider two cases a random vector $\mathbf{u} \in \mathbb{R}^d$ either from the standard normal distribution $\mathcal{N}(\mathbf{0}, \mathbf{I}_d)$ or from the Rademacher distribution with i.i.d. entries in $\{-1, +1\}$ (i.e., $\mathbb{P}(u_j = \pm 1) = \frac{1}{2}$ for all $j = 1, \dots, d$). Given $\mathbf{g}(\mathbf{x})$, the difference in variance between the two choices of random direction in (2) satisfies,*
$$Var_{\mathbf{u} \sim \mathcal{N}(\mathbf{0}, \mathbf{I}_d)}[\hat{\mathbf{g}}(\mathbf{x})] - Var_{\mathbf{u} \sim Rademacher^d}[\hat{\mathbf{g}}(\mathbf{x})] = 2\, \|\mathbf{g}(\mathbf{x})\|^2\, \mathbf{I}_d, \tag{13}$$
*where $\hat{\mathbf{g}}(\mathbf{x})$ is the projected directional derivative defined in (2).*

*Proof.* Recall the variance formula
$$\text{Var}[\hat{\mathbf{g}}(\mathbf{x})] = \mathbb{E}[\hat{\mathbf{g}}(\mathbf{x})\, \hat{\mathbf{g}}(\mathbf{x})^\top] - \mathbb{E}[\hat{\mathbf{g}}(\mathbf{x})]\, \mathbb{E}[\hat{\mathbf{g}}(\mathbf{x})]^\top. \tag{14}$$
We compute $\text{Var}[\hat{\mathbf{g}}(\mathbf{x})]$ for the cases where $\mathbf{u}$ is drawn from either a normal or rademacher distribution. First, we begin with the case where $\mathbf{u} \sim \mathcal{N}(\mathbf{0}, \mathbf{I}_d)$. Recall, for a random vector $\mathbf{u} \sim \mathcal{N}(\mathbf{0}, \mathbf{I}_d)$, we have
$$\mathbb{E}[\mathbf{u}] = \mathbf{0}, \ \ \mathbb{E}[\mathbf{u}\mathbf{u}^\top] = \mathbf{I}_d.$$
Thus, we have $\mathbb{E}[\hat{\mathbf{g}}(\mathbf{x})] = \mathbf{g}(\mathbf{x})$. Next, compute $\mathbb{E}[\hat{\mathbf{g}}(\mathbf{x})\hat{\mathbf{g}}(\mathbf{x})^\top]$ in (14)
$$\mathbb{E}[\hat{\mathbf{g}}(\mathbf{x})\hat{\mathbf{g}}(\mathbf{x})^\top] = \mathbb{E}[(\mathbf{u}^\top \mathbf{g}(\mathbf{x}))^2 \mathbf{u}\mathbf{u}^\top]. \tag{15}$$

Note that, $\mathbf{u}\mathbf{u}^\top = \sum_{m=1}^d \sum_{p=1}^d u_m u_p \mathbf{e}_m \mathbf{e}_p^\top$ and $(\mathbf{u}^\top \mathbf{g}(\mathbf{x}))^2 = \left(\sum_{l=1}^d u_l g_l(\mathbf{x})\right)^2 = \sum_{l=1}^d \sum_{n=1}^d u_l u_n g_l(\mathbf{x}) g_n(\mathbf{x})$, where $\mathbf{e}$ is the basis vector.

Plugging back into (15), we have
$$\mathbb{E}[\hat{\mathbf{g}}(\mathbf{x})\hat{\mathbf{g}}(\mathbf{x})^\top] = \mathbb{E}[\sum_{l=1}^d \sum_{n=1}^d u_l u_n g_l(\mathbf{x}) g_n(\mathbf{x}) \sum_{m=1}^d \sum_{p=1}^d u_m u_p \mathbf{e}_m \mathbf{e}_p^\top]$$
$$= \mathbb{E}[\sum_{l=1}^d \sum_{n=1}^d \sum_{m=1}^d \sum_{p=1}^d u_l u_n g_l(\mathbf{x}) g_n(\mathbf{x}) u_m u_p \mathbf{e}_m \mathbf{e}_p^\top]$$
$$= \sum_{l=1}^d \sum_{n=1}^d \sum_{m=1}^d \sum_{p=1}^d g_l(\mathbf{x}) g_n(\mathbf{x})\, \mathbb{E}[u_l u_n u_m u_p] \mathbf{e}_m \mathbf{e}_p^\top. \tag{16}$$

Note that, the last equality comes from the fact that $\mathbf{u}$ is independent from $\mathbf{g}(\mathbf{x})$. Now, there is different cases that $\mathbb{E}[u_l u_n u_m u_p]$ is non-zero in (16).

Case 1 ($l = n$ and $m = p$): In this case, (16) simplifies to the following

$$\mathbb{E}[\hat{\mathbf{g}}(\mathbf{x})\hat{\mathbf{g}}(\mathbf{x})^\top] = \sum_{l=1}^{d}\sum_{m=1}^{d} g_l^2(\mathbf{x})\,\mathbb{E}[u_l^2]\,\mathbb{E}[u_m^2]\mathbf{e}_m\mathbf{e}_m^\top = \|\mathbf{g}(\mathbf{x})\|^2\mathbf{I}_d.$$

Case 2 ($l = m$ and $n = p$): In this case, (16) simplifies to the following

$$\mathbb{E}[\hat{\mathbf{g}}(\mathbf{x})\hat{\mathbf{g}}(\mathbf{x})^\top] = \sum_{l=1}^{d}\sum_{n=1}^{d} g_l(\mathbf{x})g_n(\mathbf{x})\mathbb{E}[u_l^2]\mathbb{E}[u_n^2]\mathbf{e}_l\mathbf{e}_n^\top = \mathbf{g}(\mathbf{x})(\mathbf{g}(\mathbf{x}))^\top$$

Case 3 ($l = p$ and $m = n$): Similar to case 2, (16) simplifies to the following

$$\mathbb{E}[\hat{\mathbf{g}}(\mathbf{x})\hat{\mathbf{g}}(\mathbf{x})^\top] = \mathbf{g}(\mathbf{x})(\mathbf{g}(\mathbf{x}))^\top$$

Case 4 ($l = n = m = p$): In this case, (16) simplifies to the following

$$\mathbb{E}[\hat{\mathbf{g}}(\mathbf{x})\hat{\mathbf{g}}(\mathbf{x})^\top] = \sum_{l=1}^{d} g_l^2(\mathbf{x})\,\mathbb{E}[u_l^4]\mathbf{e}_l\mathbf{e}_l^\top = 3\|\mathbf{g}(\mathbf{x})\|^2\mathbf{I}_d$$

Then, (16) can be simplified as follows

$$\mathbb{E}[\hat{\mathbf{g}}(\mathbf{x})\hat{\mathbf{g}}(\mathbf{x})^\top] = 2\,\mathbf{g}(\mathbf{x})(\mathbf{g}(\mathbf{x}))^\top + 4\|\mathbf{g}(\mathbf{x})\|^2\mathbf{I}_d.$$

As a result, (14) can be written as follows for the case where $\mathbf{u}$ drawn from a normal distribution with zero mean and unit variance.

$$\mathrm{Var}_{\mathbf{u}\sim\mathcal{N}(\mathbf{0},\mathbf{I}_d)}[\hat{\mathbf{g}}(\mathbf{x})] = \mathbf{g}(\mathbf{x})(\mathbf{g}(\mathbf{x}))^\top + 4\|\mathbf{g}(\mathbf{x})\|^2\mathbf{I}_d. \tag{17}$$

Now, we compute the variance $\mathrm{Var}[\hat{\mathbf{g}}(\mathbf{x})]$ in the case where $\mathbf{u} \sim \mathrm{Rademacher}^d$. Recall that for a random vector $\mathbf{u}$ with i.i.d. Rademacher entries, we have $\mathbb{E}[\mathbf{u}] = \mathbf{0}$ and $\mathbb{E}[\mathbf{u}\mathbf{u}^\top] = \mathbf{I}_d$. Therefore, we have $\mathbb{E}[\hat{\mathbf{g}}(\mathbf{x})] = \mathbf{g}(\mathbf{x})$.

Similar to the proof for the normal distribution, cases $1, 2,$ and $3$ are identical. However in the fourth case, where $l = n = m = p$, since the fourth moment of a Rademacher distribution is 1, we get $\mathbb{E}[\hat{\mathbf{g}}(\mathbf{x})\hat{\mathbf{g}}(\mathbf{x})^\top] = \|\mathbf{g}(\mathbf{x})\|^2\mathbf{I}_d$. Then, (16) can be simplified as follows

$$\mathbb{E}[\hat{\mathbf{g}}(\mathbf{x})\hat{\mathbf{g}}(\mathbf{x})^\top] = 2\,\mathbf{g}(\mathbf{x})(\mathbf{g}(\mathbf{x}))^\top + 2\|\mathbf{g}(\mathbf{x})\|^2\mathbf{I}_d.$$

Thus, we get

$$\mathrm{Var}_{\mathbf{u}\sim\mathrm{Rademacher}^d}[\hat{\mathbf{g}}(\mathbf{x})] = \mathbf{g}(\mathbf{x})(\mathbf{g}(\mathbf{x}))^\top + 2\|\mathbf{g}(\mathbf{x})\|^2\mathbf{I}_d. \tag{18}$$

From (17) and (18), we have the following

$$\mathrm{Var}_{\mathbf{u}\sim\mathcal{N}(\mathbf{0},\mathbf{I}_d)}[\hat{\mathbf{g}}(\mathbf{x})] - \mathrm{Var}_{\mathbf{u}\sim\mathrm{Rademacher}^d}[\hat{\mathbf{g}}(\mathbf{x})] = 2\|\mathbf{g}(\mathbf{x})\|^2\mathbf{I}_d \tag{19}$$

which concludes the proof. $\qquad\square$

**Lemma 4** (The Non-Negativity of the Dot Product Between (2) and $\mathbf{g}(\mathbf{x})$). *For any given realization of $\mathbf{g}(\mathbf{x})$ and any vector $\mathbf{u} \in \mathbb{R}^d$, the projected directional derivative $\hat{\mathbf{g}}(\mathbf{x}) := \left(\mathbf{u}^\top\mathbf{g}(\mathbf{x})\right)\mathbf{u}$ satisfies $\mathbf{g}(\mathbf{x})^\top\hat{\mathbf{g}}(\mathbf{x}) \geq 0.$*

*Proof.* Since

$$\mathbf{g}(\mathbf{x})^\top\hat{\mathbf{g}}(\mathbf{x}) = \mathbf{g}(\mathbf{x})^\top\left(\mathbf{u}^\top\mathbf{g}(\mathbf{x})\right)\mathbf{u} = \left(\mathbf{u}^\top\mathbf{g}(\mathbf{x})\right)^2 = s^2 \geq 0,$$

where $s = \mathbf{u}^\top\mathbf{g}(\mathbf{x}) \in \mathbb{R}$, we conclude that the projection $\hat{\mathbf{g}}(\mathbf{x})$ forms a non-negative inner product with $\mathbf{g}(\mathbf{x})$. Therefore, for every $\mathbf{u} \in \mathbb{R}^d$, the estimate $\hat{\mathbf{g}}(\mathbf{x})$ is aligned with $\mathbf{g}(\mathbf{x})$. $\qquad\square$

**Lemma 5** (Bounded Diversity of $f_i$, T Dinh et al. (2020), Lemma 4)**.** *Let $\mathcal{A}_k \subseteq \{1, \ldots, N\}$ denote the random set of clients selected at round $k$ with size $|\mathcal{A}_k| = \beta N$, where $\beta \in (0, 1]$. Then, we have*

$$\mathbb{E}_{\mathcal{A}_k} \left\| \frac{1}{\beta N} \sum_{i \in \mathcal{A}_k} \nabla f_i(\mathbf{x}_k) - \nabla f(\mathbf{x}_k) \right\|_2^2 \leq \frac{1/\beta - 1}{N - 1} \cdot \frac{1}{N} \sum_{i=1}^{N} \|\nabla f_i(\mathbf{x}_k) - \nabla f(\mathbf{x}_k)\|_2^2.$$

$\square$

**Lemma 6** (Johnson–Lindenstrauss Lemma Rudelson & Vershynin (2008); Tropp (2012))**.** Fix $0 < \varepsilon < 1$ and $0 < \delta < 1$. Let $d \in \mathbb{N}$ be the ambient dimension and choose an embedding dimension $m = O\big(\frac{\ln(d/\delta)}{\varepsilon^2}\big)$. Draw a random matrix $P \in \mathbb{R}^{m \times d}$ whose entries are i.i.d. mean-zero, unit-variance, sub-Gaussian[1] variables. Then, with probability at least $1 - \delta$,

$$(1 - \varepsilon)\|x\|^2 \leq \|Px\|^2 \leq (1 + \varepsilon)\|x\|^2, \qquad \text{for all } x \in \mathbb{R}^d.$$

Equivalently, $\left\|P^\top P - I_d\right\|_2 \leq \varepsilon$ with probability at least $1 - \delta$.

**Theorem 1** (Convergence Bound of `FedPDD` Algorithm)**.** *Consider `FedPDD` algorithm with a step size $\eta = \frac{1}{L\sqrt{K}}$, and suppose that Assumption 1 holds. Then, `FedPDD` algorithm converges to a stationary point of problem (1) at a rate of $O(d/\sqrt{K})$, satisfying the following upper bound*

$$\frac{1}{K} \sum\nolimits_{k=0}^{K-1} \mathbb{E}\left[\|\nabla f(\mathbf{x}_k)\|^2\right] \leq \underbrace{O\left(\frac{L(f(\mathbf{x}_0) - f^\star)}{K^{0.5}}\right)}_{\text{due to initialization}} + \underbrace{O\left(\frac{\sigma^2(1/\beta - 1)}{K^{1.5}}\right)}_{\text{due to client sampling}} + \underbrace{O\left(\frac{dG^2}{K^{0.5}}\right)}_{\text{due to projected directional derivative}}, \qquad (20)$$

*where $\beta \in (0, 1]$ denotes the client participation fraction, and $f^\star$ denotes the global minimum of $f$.*

*Proof.* Since $f$ is $L - smooth$, we can write

$$\mathbb{E}[f(\mathbf{x}_{k+1})|\mathbf{x}_k] \leq f(\mathbf{x}_k) + \left\langle \nabla f(\mathbf{x}_k), \mathbb{E}[\mathbf{x}_{k+1} - \mathbf{x}_k|\mathbf{x}_k] \right\rangle + \frac{L}{2} \mathbb{E}\left[\|\mathbf{x}_{k+1} - \mathbf{x}_x\|^2 \big| \mathbf{x}_k\right], \qquad (21)$$

from line 14 of `FedPDD` algorithm into (21), we have

$$\mathbb{E}[f(\mathbf{x}_{k+1})|\mathbf{x}_k] \leq f(\mathbf{x}_k) - \eta \left\langle \nabla f(\mathbf{x}_k), \mathbb{E}[\hat{\mathbf{g}}(\mathbf{x}_k)|\mathbf{x}_k] \right\rangle + \frac{L\eta^2}{2} \mathbb{E}\left[\|\hat{\mathbf{g}}(\mathbf{x}_k)\|^2 \big| \mathbf{x}_k\right],$$

$$= f(\mathbf{x}_k) - \eta \|\nabla f(\mathbf{x}_k)\|^2 + \frac{L\eta^2}{2} \mathbb{E}\left[\|\hat{\mathbf{g}}(\mathbf{x}_k)\|^2 \big| \mathbf{x}_k\right],$$

$$= f(\mathbf{x}_k) - \eta \|\nabla f(\mathbf{x}_k)\|^2 + \frac{L\eta^2}{2} \left( \mathbb{E}\left[\|\hat{\mathbf{g}}(\mathbf{x}_k) - \nabla f(\mathbf{x}_k)\|^2 \big| \mathbf{x}_k\right] + \|\nabla f(\mathbf{x}_k)\|^2 \right),$$

$$= f(\mathbf{x}_k) - (\eta - \frac{L\eta^2}{2}) \|\nabla f(\mathbf{x}_k)\|^2 + \frac{L\eta^2}{2} \mathbb{E}\left[\|\hat{\mathbf{g}}(\mathbf{x}_k) - \nabla f(\mathbf{x}_k)\|^2 \big| \mathbf{x}_k\right],$$

$$\qquad (22)$$

where the first equality comes from the unbiasedness of $\hat{\mathbf{g}}(\mathbf{x}_k)$, i.e., $\mathbb{E}[\hat{\mathbf{g}}(\mathbf{x}_k)|\mathbf{x}_k] = \nabla f(\mathbf{x}_k)$, the second inequality comes from bias–variance decomposition of $\mathbb{E}[\|\hat{\mathbf{g}}(\mathbf{x}_k)\|^2|\mathbf{x}_k]$. Provided that $\eta < \frac{1}{L}$, and taking total expectation on both sides of (22) we have

$$\mathbb{E}[f(\mathbf{x}_{k+1})] \leq \mathbb{E}[f(\mathbf{x}_k)] - \frac{\eta}{2} \mathbb{E}[\|\nabla f(\mathbf{x}_k)\|^2] + \frac{L\eta^2}{2} \underbrace{\mathbb{E}\left[\|\hat{\mathbf{g}}(\mathbf{x}_k) - \nabla f(\mathbf{x}_k)\|^2\right]}_{A},$$

$$\qquad (23)$$

---

[1]For example, take $P_{ij} = \pm 1/\sqrt{k}$ with probability $1/2$ each, or $P_{ij} \sim \mathcal{N}(0, 1/k)$.

First we simplify $A$ in (23) by adding and subtracting $\frac{1}{\beta N} \sum_{i \in \mathcal{A}_k} \mathbf{g}_i(\mathbf{x}_k)$

$$
\begin{aligned}
\mathbb{E}\left[\|\hat{\mathbf{g}}(\mathbf{x}_k) - \nabla f(\mathbf{x}_k)\|^2\right] &= \mathbb{E}\left[\|\frac{1}{\beta N} \sum_{i \in \mathcal{A}_k} \mathbf{g}_i(\mathbf{x}_k) - \nabla f(\mathbf{x}_k) + \hat{\mathbf{g}}(\mathbf{x}_k) - \frac{1}{\beta N} \sum_{i \in \mathcal{A}_k} \mathbf{g}_i(\mathbf{x}_k)\|^2\right] \\
&\leq 2\,\mathbb{E}\left[\|\frac{1}{\beta N} \sum_{i \in \mathcal{A}_k} \mathbf{g}_i(\mathbf{x}_k) - \nabla f(\mathbf{x}_k)\|^2\right] + 2\,\mathbb{E}[\|\hat{\mathbf{g}}(\mathbf{x}_k) - \frac{1}{\beta N} \sum_{i \in \mathcal{A}_k} \mathbf{g}_i(\mathbf{x}_k)\|^2] \\
&= 2\,\mathbb{E}\left[\|\frac{1}{\beta N} \sum_{i \in \mathcal{A}_k} \mathbf{g}_i(\mathbf{x}_k) - \nabla f(\mathbf{x}_k)\|^2\right] + 2\,\mathbb{E}[\|\frac{1}{\beta N} \sum_{i \in \mathcal{A}_k} s_i^k \mathbf{u}_{k,i} - \frac{1}{\beta N} \sum_{i \in \mathcal{A}_k} \mathbf{g}_i(\mathbf{x}_k)\|^2] \\
&\leq 2\,\mathbb{E}\left[\|\frac{1}{\beta N} \sum_{i \in \mathcal{A}_k} \mathbf{g}_i(\mathbf{x}_k) - \nabla f(\mathbf{x}_k)\|^2\right] + \frac{2}{\beta N} \sum_{i \in \mathcal{A}_k} \mathbb{E}[\|s_i^k \mathbf{u}_{k,i} - \mathbf{g}_i(\mathbf{x}_k)\|^2],
\end{aligned}
\tag{24}
$$

where the first inequality follows from $\|a + b\|^2 \leq 2\|a\|^2 + 2\|b\|^2$, and the second inequality follows from Jensen's inequality. Using Lemma 5 into (24) we get

$$
\begin{aligned}
\mathbb{E}\left[\|\hat{\mathbf{g}}(\mathbf{x}_k) - \nabla f(\mathbf{x}_k)\|^2\right] &\leq \frac{2(1/\beta - 1)}{N(N-1)} \sum_{i=1}^{N} \mathbb{E}[\|\nabla f_i(\mathbf{x}_k) - \nabla f(\mathbf{x}_k)\|^2]) \\
&\quad + \frac{2}{\beta N} \sum_{i \in \mathcal{A}_k} \mathbb{E}[\|s_i^k \mathbf{u}_{k,i} - \mathbf{g}_i(\mathbf{x}_k)\|^2], \\
&= \frac{2(1/\beta - 1)}{N(N-1)} \sum_{i=1}^{N} \mathbb{E}[\|\nabla f_i(\mathbf{x}_k) - \nabla f(\mathbf{x}_k)\|^2]) + \frac{2}{\beta N} \sum_{i \in \mathcal{A}_k} \mathbb{E}[\|\hat{\mathbf{g}}_i(\mathbf{x}_k) - \mathbf{g}_i(\mathbf{x}_k)\|^2],
\end{aligned}
\tag{25}
$$

where the equality comes from the fact that $s_i^k \mathbf{u}_{k,i} = \hat{\mathbf{g}}_i(\mathbf{x}_k)$. Now, using the result from Lemma **??** in (25) we get

$$
\begin{aligned}
\mathbb{E}\left[\|\hat{\mathbf{g}}(\mathbf{x}_k) - \nabla f(\mathbf{x}_k)\|^2\right] &\leq \frac{2(1/\beta - 1)}{N(N-1)} \sum_{i=1}^{N} \mathbb{E}[\|\nabla f_i(\mathbf{x}_k) - \nabla f(\mathbf{x}_k)\|^2]) \\
&\quad + \frac{2(d-1)}{\beta N} \sum_{i \in \mathcal{A}_k} \mathbb{E}[\|\mathbf{g}_i(\mathbf{x}_k)\|^2].
\end{aligned}
\tag{26}
$$

Using (26) into (23) we have

$$
\begin{aligned}
\mathbb{E}[f(\mathbf{x}_{k+1})] &\leq \mathbb{E}[f(\mathbf{x}_k)] - \frac{\eta}{2}\,\mathbb{E}[\|\nabla f(\mathbf{x}_k)\|^2] \\
&\quad + \frac{L\eta^2}{2}\left(\frac{2(1/\beta - 1)}{N(N-1)} \sum_{i=1}^{N} \mathbb{E}[\|\nabla f_i(\mathbf{x}_k) - \nabla f(\mathbf{x}_k)\|^2]) + \frac{2(d-1)}{\beta N} \sum_{i \in \mathcal{A}_k} \mathbb{E}[\|\mathbf{g}_i(\mathbf{x}_k)\|^2]\right).
\end{aligned}
\tag{27}
$$

Invoking Assumption 1, which bounds the variance of local gradients relative to the global gradient in (27), we get

$$
\begin{aligned}
\mathbb{E}[f(\mathbf{x}_{k+1})] &\leq \mathbb{E}[f(\mathbf{x}_k)] - \frac{\eta}{2}\,\mathbb{E}[\|\nabla f(\mathbf{x}_k)\|^2] \\
&\quad + \frac{L\eta^2}{2}\left(\frac{2(1/\beta - 1)}{N-1}\sigma^2 + \frac{2(d-1)}{\beta N} \sum_{i \in \mathcal{A}_k} \mathbb{E}[\|\mathbf{g}_i(\mathbf{x}_k)\|^2]\right).
\end{aligned}
\tag{28}
$$

Rearranging and summing $k$ from 0 to $K-1$ in (28), we have

$$\frac{1}{K} \sum_{k=0}^{K-1} \mathbb{E}[\|\nabla f(\mathbf{x}_k)\|^2] \leq \frac{2}{K\eta}(\mathbb{E}[f(\mathbf{x}_0)] - \mathbb{E}[f(\mathbf{x}_K)]) + \frac{2L\eta(1/\beta - 1)}{K(N-1)}\sigma^2$$

$$+ \frac{2L\eta(d-1)}{K\beta N} \sum_{k=0}^{K-1} \sum_{i \in \mathcal{A}_k} \mathbb{E}\left[\|\mathbf{g}_i(\mathbf{x}_k)\|^2\right]$$

$$\leq \frac{2}{K\eta}(\mathbb{E}[f(\mathbf{x}_0)] - f^\star) + \frac{2L\eta(1/\beta - 1)}{K(N-1)}\sigma^2 + \frac{2L\eta(d-1)}{K\beta N} \sum_{k=0}^{K-1} \sum_{i \in \mathcal{A}_k} \mathbb{E}\left[\|\mathbf{g}_i(\mathbf{x}_k)\|^2\right]$$

$$(29)$$

We simplify the upper bound in (29) by invoking Assumption 1 on the bounded variance of the stochastic gradient, we get

$$\frac{1}{K} \sum_{k=0}^{K-1} \mathbb{E}[\|\nabla f(\mathbf{x}_k)\|^2] \leq \frac{2}{K\eta}(f(\mathbf{x}_0) - f^\star) + \frac{2L\eta(1/\beta - 1)}{K(N-1)}\sigma^2 + 2L\eta d G^2. \quad (30)$$

If we set $\eta = \frac{1}{L\sqrt{K}}$ in (30), we have a convergence rate of $O(d/\sqrt{K})$ to a stationary point of $f(\mathbf{x})$. $\qquad \square$

**Theorem 2** (Convergence Bound of `FedMPDD` Algorithm). *Let the step size $\eta = \frac{1}{L\sqrt{K}}$, and suppose that Assumption 1 holds. Let number of random vectors be $m = O\left(\frac{\ln(d/\delta)}{\varepsilon^2}\right)$. Then, `FedMPDD` algorithm converges to a stationary point of problem (1) at a rate of $O(1/\sqrt{K})$, satisfying the following upper bound with probability at least $1 - \delta$,*

$$\frac{1}{K} \sum_{k=0}^{K-1} \mathbb{E}\left[\|\nabla f(\mathbf{x}_k)\|^2\right] \leq \underbrace{O\left(\frac{L(f(\mathbf{x}_0) - f^\star)}{K^{0.5}}\right)}_{\text{due to initialization}} + \underbrace{O\left(\frac{\sigma^2(1/\beta - 1)}{K^{1.5}}\right)}_{\text{due to client sampling}} + \underbrace{O\left(\frac{\epsilon G^2}{K^{0.5}}\right)}_{\text{due to Multi-projected directional derivatives}} , \quad (31)$$

*where $0 < \epsilon < 1$ is the distortion parameter, $\beta \in (0, 1]$ denotes the client participation fraction, and $f^\star$ denotes the global minimum of $f$.*

*Proof.* For notational convenience, we stack the $m$ random direction vectors into a single matrix $U_{k,i} = \begin{bmatrix} \mathbf{u}_{k,i}^{(1)} & \mathbf{u}_{k,i}^{(2)} & \dots & \mathbf{u}_{k,i}^{(m)} \end{bmatrix} \in \mathbb{R}^{d \times m}$, which simplifies the analysis that follows. With this compact form, the gradient estimator in `FedMPDD` algorithm $\hat{\mathbf{g}}_i(\mathbf{x}_k) = \frac{1}{m} U_{k,i} U_{k,i}^\top \mathbf{g}_i(\mathbf{x}_k)$ remains unbiased, i.e., $\mathbb{E}[\hat{\mathbf{g}}_i(\mathbf{x}_k)] = \frac{1}{m}\mathbb{E}\left[U_{k,i} U_{k,i}^\top\right] \mathbf{g}_i(\mathbf{x}_k) = \mathbf{g}_i(\mathbf{x}_k)$, most of the derivation steps leading up to Equation (25) remain unchanged. Therefore, we omit those details for brevity. From (25) in the proof of Theorem 1, we have

$$\mathbb{E}\left[\|\hat{\mathbf{g}}(\mathbf{x}_k) - \nabla f(\mathbf{x}_k)\|^2\right] \leq \frac{2(1/\beta - 1)}{N(N-1)} \sum_{i=1}^{N} \mathbb{E}[\|\nabla f_i(\mathbf{x}_k) - \nabla f(\mathbf{x}_k)\|^2]$$

$$+ \frac{2}{\beta N} \sum_{i \in \mathcal{A}_k} \mathbb{E}[\|\hat{\mathbf{g}}_i(\mathbf{x}_k) - \mathbf{g}_i(\mathbf{x}_k)\|^2],$$

$$= \frac{2(1/\beta - 1)}{N(N-1)} \sum_{i=1}^{N} \mathbb{E}[\|\nabla f_i(\mathbf{x}_k) - \nabla f(\mathbf{x}_k)\|^2]$$

$$+ \frac{2}{\beta N} \sum_{i \in \mathcal{A}_k} \mathbb{E}[\|\hat{\mathbf{g}}_i(\mathbf{x}_k)\|^2] - \mathbb{E}[\|\mathbf{g}_i(\mathbf{x}_k)\|^2],$$

$$(32)$$

the last equality holds because the estimator is unbiased and the Rademacher directions are sampled independently of all other sources of randomness. Where $\hat{\mathbf{g}}(\mathbf{x}_k) = \frac{1}{m\beta N} \sum_{i \in \mathcal{A}_k} U_{k,i} U_{k,i}^\top \mathbf{g}_i(\mathbf{x}_k)$, with $\beta N = |\mathcal{A}_k|$ denoting the number of participating clients at round $k$, and $\beta \in (0, 1]$ representing

the client sampling fraction. Using JL Lemma 6 results, for $m = O\left(\frac{\ln(d/\delta)}{\varepsilon^2}\right)$ into (32) with probability at least $1 - \delta$, we have

$$\mathbb{E}\left[\|\hat{\mathbf{g}}(\mathbf{x}_k) - \nabla f(\mathbf{x}_k)\|^2\right] \leq \frac{2(1/\beta - 1)}{N(N-1)} \sum_{i=1}^{N} \mathbb{E}[\|\nabla f_i(\mathbf{x}_k) - \nabla f(\mathbf{x}_k)\|^2]$$

$$+ \frac{\epsilon(4 + 2\epsilon)}{\beta N} \sum_{i \in \mathcal{A}_k} \mathbb{E}[\|\mathbf{g}_i(\mathbf{x}_k)\|^2].$$

(33)

Now, similar to the proof steps of Theorem 1 following the steps after equation (25) and invoking Assumption 1, we obtain the following upper bound

$$\frac{1}{K} \sum_{k=0}^{K-1} \mathbb{E}[\|\nabla f(\mathbf{x}_k)\|^2] \leq \frac{2}{K\eta}(f(\mathbf{x}_0) - f^\star) + \frac{2L\eta(1/\beta - 1)}{K(N-1)}\sigma^2 + \epsilon(4 + 2\epsilon)L\eta G^2, \quad (34)$$

where $0 < \epsilon < 1$ is the distortion parameter. If we set $\eta = \frac{1}{L\sqrt{K}}$ in (34), we have a convergence rate of $O(1/\sqrt{K})$ to a stationary point of $f(\mathbf{x})$.

$\square$

| Method | Time complexity | Memory complexity |
|---|---|---|
| Backprop | $O(h^2 T + hpT)$ | $O((h+p)T)$ |
| Forwardprop (full JVP) | $O(h^2 p T^2)$ | $O(h+p)$ |
| Projected-Forward | $O(h^2 T + hpT)$ | $O(h+p)$ |

Table F.1: Time and memory complexities of gradient computation methods in a depth-$T$ neural network with hidden dimension $h$ and $p$ parameters per layer.

---

*Instead of computing $\mathbf{g}_i(\mathbf{x}_k)$, the same mini-batch $\mathcal{B}_i^k$ is reused across all random directions.*
1: **for** $j = 1, \ldots, m$ **do**
2:     Generate i.i.d. Rademacher vector $\mathbf{u}_k^{(j)} \in \{-1, +1\}^d$ using seed $r_k$ and index $j$
3:     $\mathbf{s}_i^k[j] \leftarrow \mathrm{JVP}\big(f_i, \mathbf{x}_k, \mathbf{u}_k^{(j)}; \mathcal{B}_i^k\big)$
4: **end for**
5: Upload $\mathbf{s}_i^k \in \mathbb{R}^m$

---

Figure F.1: Potential usage of `FedMPDD` for faster gradient computation in deep neural networks.

## F  DISCUSSION AND FUTURE WORK

As noted in Remark 1, an important aspect of `FedMPDD` is the potential computational cost introduced on the client side. To mitigate this, `FedMPDD` can adopt the projected-forward approach illustrated in Figure F.1, where a fixed mini-batch is sampled once and reused to compute $m$ directional derivatives via forward-mode Jacobian-vector products. When $m < \frac{hpT}{h+p}$, this approach not only reduces communication overhead and enhances privacy, but also lowers computational cost on the client side (a crucial consideration in federated learning, where participating devices often have limited compute and memory resources). Standard methods for gradient computation, such as backpropagation and full forward-mode differentiation, typically incur high memory or time complexity. In contrast, the projected-forward approach provides a balance between these two challenges.

Motivated by the promising initial results of the projected-forward approach for gradient computation in deep neural networks, we plan to implement a fully optimized version of `FedMPDD` using Jacobian-vector products (JVPs), which leverage efficient vector–matrix multiplications. This implementation will `FedMPDD` to simultaneously enhance privacy and communication efficiency while reducing the computational time complexity on the client side. On the theoretical side, since the *projected directional derivative* is an unbiased estimator, it offers an opportunity to further accelerate convergence by incorporating momentum and variance reduction techniques into `FedMPDD`.

---

**Algorithm 1** FedPDD: **Fed**erated Learning via **P**rojected **D**irectional **D**erivative

---

1: **Input:** $\mathbf{x}_0 \in \mathbb{R}^d$, learning rate $\eta$, rounds $K$, client fraction $\beta \in (0, 1]$
2: **for** each round $k = 0, 1, \ldots, K-1$ **do**
3:      Server samples client set $\mathcal{A}_k$ with size $\beta N$
4:      Server broadcasts $\mathbf{x}_k$ to all $i \in \mathcal{A}$
5:      **for** each client $i \in \mathcal{A}_k$ **in parallel do**
6:          Generate i.i.d. Rademacher vector $\mathbf{u}_k \in \{-1, +1\}^d$ using seed $r_{k,i}$
7:          Compute local stochastic gradient $\mathbf{g}_i(\mathbf{x}_k)$
8:          **Encode:** $s_i^k = \mathbf{u}_{k,i}^\top \mathbf{g}_i(\mathbf{x}_k)$
9:          Upload $s_i^k \in \mathbb{R}$ and $r_{k,i} \in \mathbb{R}$ to the server
10:      **end for**

11:      $\boldsymbol{\Delta}_{\text{sum}} \leftarrow \mathbf{0}_d$                            $\triangleright$ reset the estimator
12:      **for** each client $i \in \mathcal{A}_k$ **do**                  $\triangleright$ on the server side
13:          Server generates Rademacher vector $\mathbf{u}_{k,i} \in \{-1, +1\}^d$ using seed $r_{k,i}$
14:          **Decode:** $\boldsymbol{\Delta}_{\text{sum}} \leftarrow \boldsymbol{\Delta}_{\text{sum}} + s_i^k \mathbf{u}_{k,i}$
15:      **end for**
16:      **Aggregate:** $\hat{\mathbf{g}}(\mathbf{x}_k) = \dfrac{1}{\beta N} \boldsymbol{\Delta}_{\text{sum}}$
17:      **Model update:** $\mathbf{x}_{k+1} = \mathbf{x}_k - \eta \hat{\mathbf{g}}(\mathbf{x}_k)$
18: **end for**
19: **Output:** $\mathbf{x}_K$

---

# G    ALGORITHM 1: FEDERATED LEARNING VIA PROJECTED DIRECTIONAL DERIVATIVE (FEDPDD)

The FedPDD algorithm is presented in detail in Algorithm 1.

Table H.1: Model architecture of LeNet.

| Layer | Type | Kernel Size | Output Shape | Activation |
|-------|------|-------------|--------------|------------|
| Input | - | - | $1 \times 28 \times 28$ | - |
| Conv1 | Conv2D | $5 \times 5$, stride 2, padding 2 | $12 \times 14 \times 14$ | Sigmoid |
| Conv2 | Conv2D | $5 \times 5$, stride 2, padding 2 | $12 \times 7 \times 7$ | Sigmoid |
| Conv3 | Conv2D | $5 \times 5$, stride 1, padding 2 | $12 \times 7 \times 7$ | Sigmoid |
| Flatten | - | - | 588 | - |
| FC | Linear | - | 10 (logits) | - |

Table H.2: Model Architecture of CNN in (Lin et al., 2022).

| Layer | Type | Kernel / Params | Output Shape | Activation |
|-------|------|-----------------|--------------|------------|
| Input | - | - | $1 \times 28 \times 28$ | - |
| Conv1 | Conv2D | $5 \times 5$, MaxPool(2) | $6 \times 12 \times 12$ | ReLU |
| Conv2 | Conv2D | $5 \times 5$, MaxPool(2) | $16 \times 4 \times 4$ | ReLU |
| Flatten | - | - | 400 | - |
| FC1 | Linear | $400 \rightarrow 120$ | 120 | ReLU |
| FC2 | Linear | $120 \rightarrow 84$ | 84 | ReLU |
| FC3 | Linear | $84 \rightarrow 10$ | 10 (logits) | - |

## H  REPRODUCIBILITY AND EXPERIMENTAL SETUP

### H.1  SUMMARY OF MODELS AND DATASETS

To ensure a fair and comprehensive evaluation of the proposed algorithm against the baseline methods, we conduct experiments using multiple neural network architectures with varying parameter sizes. Unless stated otherwise, data is partitioned among 100 clients. First, we describe the model architectures used in our experiments below.

- **LeNet for MNIST and FASHIONMNIST:** The architecture of the LeNet model used for these datasets is detailed in Table H.1.
- **CNN from Lin et al. (2022) for MNIST and FASHIONMNIST:** The model architecture is adapted for the MNIST and FASHIONMNIST datasets and is detailed in Table H.2.
- **CNN from McMahan et al. (2017) for CIFAR10:** The architecture of the model used for the dataset is detailed in Table H.3.

Note that, MNIST and FASHIONMNIST datasets each contain 60,000 training samples and 10,000 test samples. The CIFAR-10 dataset consists of 50,000 training samples and 10,000 test samples. Moreover, under the non-IID data distribution, each client receives data from exactly two classes in the dataset.

Table H.3: Model Architecture of CNN in (McMahan et al., 2017).

| Layer | Type | Kernel / Params | Output Shape | Activation |
|-------|------|-----------------|--------------|------------|
| Input | - | - | $3 \times 32 \times 32$ | - |
| Conv1 | Conv2D | $3 \times 3$ | $128 \times 30 \times 30$ | ReLU |
|  | MaxPool | $2 \times 2$ | $128 \times 15 \times 15$ | - |
| Conv2 | Conv2D | $3 \times 3$ | $128 \times 13 \times 13$ | ReLU |
|  | MaxPool | $2 \times 2$ | $128 \times 6 \times 6$ | - |
| Conv3 | Conv2D | $3 \times 3$ | $128 \times 4 \times 4$ | ReLU |
| Flatten | - | - | 2048 | - |
| FC1 | Linear | $2048 \rightarrow 10$ | 10 (logits) | - |

Table H.4: Hyperparameters used for all methods on the logistic regression model.

| Method | batch | lr | opt. | client participation (%) |
|---|---|---|---|---|
| FedSGD | 1 | 0.01 | sgd | 10 |
| QSGD | 1 | 0.01 | sgd | 10 |
| **FedMPDD** | 1 | 0.01 | sgd | 10 |
| FedSGD + Gaussian | 1 | 0.01 | sgd | 10 |
| FedSGD + Laplace | 1 | 0.01 | sgd | 10 |

Table H.5: Hyperparameters used for all methods on the CNN model from (Lin et al., 2022).

| Method | batch | lr | opt. | client participation (%) |
|---|---|---|---|---|
| FedSGD | 64 | 0.1 | sgd | 50 |
| QSGD | 64 | 0.1 | sgd | 50 |
| **FedMPDD** | 64 | 0.1 | sgd | 50 |
| FedSGD + Laplace | 64 | 0.1 | sgd | 50 |

## H.2 HYPERPARAMETERS

In this section, we summarize the hyperparameters used for each baseline method and our proposed algorithm. To ensure reproducibility, we selected five fixed random seeds as $[17, 123, 777, 2023, 424242]$. Additionally, we use a separate seed, $2024$, to control client data partitioning.

- **Logistic model on the MNIST dataset.** See Table H.4. The best stepsize for each algorithm is selected from the set $\{0.1, 0.01, 0.001\}$.

- **CNN model from (Lin et al., 2022) on the MNIST and FASHIONMNIST datasets.** See Table H.5. The best stepsize for each algorithm is selected from the set $\{0.1, 0.01, 0.001\}$.

- **CNN model from (McMahan et al., 2017) on CIFAR10 dataset.** See Table H.6. The best stepsize for each algorithm is selected from the set $\{0.01, 0.005, 0.0001\}$.

- **LeNet model on the MNIST and FASHIONMNIST datasets.** See Table H.7. The best stepsize for each algorithm is selected from the set $\{0.1, 0.01, 0.001\}$.

For the DLG attack hyper-parameters, we followed the procedure in (Zhu et al., 2019). To improve stability, we used the Adam optimizer for both the logistic-regression model and the CNN from (McMahan et al., 2017), and the L-BFGS optimizer (history size 100, max 20 iterations) for the LeNet model and the CNN from (Lin et al., 2022).

Table H.6: Hyperparameters used for all methods on the CNN model from (McMahan et al., 2017).

| Method | batch | lr | opt. | client participation (%) |
|---|---|---|---|---|
| FedSGD | 64 | 0.005 | sgd | 100 |
| QSGD | 64 | 0.005 | sgd | 100 |
| **FedMPDD** | 64 | 0.005 | sgd | 100 |
| FedSGD + Laplace | 64 | 0.005 | sgd | 100 |
| Top-k | 64 | 0.005 | sgd | 100 |
| lp-proj | 64 | 0.005 | sgd | 100 |
| SA-FedLora | 64 | 0.005 | sgd | 100 |

## H.3 LICENSE INFORMATION FOR DATASETS

**CIFAR10.** The original CIFAR10 dataset is available under the MIT license.

**MNIST.** The original MNIST dataset is available under the CC BY-SA 3.0 license.

**FASHIONMNIST.** The original FASHIONMNIST dataset is available under the MIT license.

Table H.7: Hyperparameters used for all methods on the LeNet model.

| Method | batch | lr | opt. | client participation (%) |
|---|---|---|---|---|
| FedSGD | 1 | 0.1 | sgd | 50 |
| QSGD | 1 | 0.1 | sgd | 50 |
| **FedMPDD** | 1 | 0.1 | sgd | 50 |
| FedSGD + Laplace | 1 | 0.1 | sgd | 50 |
| Top-k | 1 | 0.1 | sgd | 50 |
| lp-proj | 1 | 0.1 | sgd | 50 |

## H.4 HARDWARE

Experiments involving logistic regression are conducted on a MacBook Pro CPU. All other experiments are executed on an NVIDIA A100 GPU.

## I EXPLANATION OF THE DEEP LEAKAGE FROM GRADIENTS (DLG) ALGORITHM IN (ZHU ET AL., 2019)

**Deep Leakage from Gradients (DLG).** DLG (Zhu et al., 2019) is an optimization–based attack that reconstructs a client's private training example from the observed gradient that the client uploads to the server.

Let $\mathbf{g}_i(\mathbf{x})$ be the gradient with respect to the model weights $\mathbf{x}$, computed by a single client on an *unknown* data pair $(\mathbf{v}, y)$. The attacker initializes dummy inputs and labels $\mathbf{v}_1' \sim \mathcal{N}(0, 1)$ and $y_1' \sim \mathcal{N}(0, 1)$, and then iteratively refines them so that the gradient they induce matches the observed gradient $\mathbf{g}_i(\mathbf{x})$. DLG algorithm basically involves 3 steps as follow:

1. ***Dummy Gradient.*** At iteration $i$, the attacker computes the dummy gradient $\mathbf{g}_i'(\mathbf{x}) = \partial f\big(F(\mathbf{v}_i'; \mathbf{x}), \, y_i'\big)/\partial\mathbf{x}$, where $F$ is the model and $f$ the loss function (e.g., cross-entropy).

2. ***Objective.*** The discrepancy $D_i = \big\|\mathbf{g}_i'(\mathbf{x}) - \mathbf{g}_i(\mathbf{x})\big\|_2^2$ serves as a differentiable objective that measures how closely the dummy data reproduce the observed gradient.

3. ***Dummy-data update.*** Using gradient-based optimization (commonly Adam or L-BFGS), the attacker updates the dummy variables:

$$\mathbf{v}_{i+1}' = \mathbf{v}_i' - \alpha\,\nabla_{\mathbf{v}'} D_i, \quad y_{i+1}' = y_i' - \alpha\,\nabla_{y'} D_i,$$

where $\alpha$ is the step size.

After several hundred to a few thousand iterations, the optimized pair $(\mathbf{v}', y')$ closely approximates the original input that produced the observed gradient $\mathbf{g}_i(\mathbf{x})$. Consequently, the more noise present in the observed gradient, the less accurately the original input and label can be reconstructed. DLG thereby reveals a fundamental privacy risk in FedSGD and motivates the need for privacy-preserving methods such as gradient perturbation, or our proposed `FedMPDD`.

## J LLM USAGE

An LLM was used exclusively for writing purposes, specifically to polish the language and improve readability of the paper.

