# OpenReview forum: "Communication-Efficient and Private Federated Learning via Projected Directional Derivative"
_ICLR.cc/2026/Conference — ICLR 2026 Conference Withdrawn Submission_

### Official Review · Reviewer_xtn9 · 2025-10-21

**Soundness:** 3
**Presentation:** 3
**Contribution:** 3
**Rating:** 4
**Confidence:** 3

**Summary:**

The paper proposes FedMPDD, where each client communicates $m$ directional derivatives, the inner products of its local gradient with server-known random Rademacher directions, along with a seed that allows the server to regenerate those directions and form an unbiased projected-gradient estimator $(\hat g=\tfrac1m U(U^\top g))$. The paper then demonstrates that this reduces the uplink communication cost per round from $\mathcal{O}(d)$ to $\mathcal{O}(m)$. The paper also empirically reports resistance to gradient-inversion attacks and proves a convergence rate comparable to FedSGD.

**Strengths:**

- The paper addresses two important areas of federated learning: communication efficiency and privacy.
- The paper shows empirically that the method is robust to privacy attacks.

**Weaknesses:**

The paper would benefit from several clarifications to be suitable for publication at ICLR.

-  **Privacy of FedMPDD is not mathematically guaranteed.** The privacy aspect is unclear, and a precise mathematical guarantee would be helpful. As written, FedMPDD is not differentially private; it provides single-round obfuscation via low-rank projection (a deterministic linear map known to the server), rather than a formal DP guarantee. Moreover, over multiple rounds there appears to be a risk of recovery. If gradients change only slightly across rounds (e.g., due to Lipschitz continuity in the parameters or repeated batches), one can model $g_t \approx g + e_t$ with small drift. Stacking the observations across $T$ rounds yields a linear system of the form $Y = A g + \eta$, where $Y$ concatenates $\{y_t\}$ and $A$ stacks $\{U_t^\top\}$. When the union of subspaces spans $\mathbb{R}^d$ (plausible once $mT \ge d$ with generic/random directions), $A$ has full column rank and the server can solve a least-squares problem to approximate $g$. This seems to enable approximate gradient (and potentially input) reconstruction over time. If this reasoning is incorrect, clarification of the adversary model and a formal guarantee (DP, information-theoretic, or otherwise) that rules out such multi-round accumulation would be valuable.

- **Convergence wording mismatch.** The formal analysis supports $\mathcal{O}(1/\sqrt K)$, whereas elsewhere (abstract/prose) it is described as $\mathcal{O}(1/K)$. Consistent wording would improve clarity.

- **Unconstrained performance/saturation.** The fixed-budget results are useful, but it would be beneficial to also show saturation (no byte cap) to make the maximum achievable accuracy clear and to quantify the gap to FedSGD at convergence. If the saturation accuracy under FedMPDD is substantially lower, this raises scalability concerns. For example, the current results report only ≈77% on MNIST/LeNet in a very simple setting; it would be informative to see whether FedMPDD can achieve higher accuracy without any constraints (FedSGD would be expected to do so).

- **Missing close baselines (TOFU, DOSFL).** These [1, 2] target similar goals (communication efficiency with privacy benefits). Comparative results and citations would help position the contribution.

- **Non-IID coverage is narrow + missing statistics.** Non-IID appears to be a class-shard split (e.g., two classes per client). Many recent FL works also report Dirichlet-$(\alpha)$ sweeps across skews (e.g., $(\alpha\in\{0.1,0.3,1.0,10.0\})$). A precise statement of the non-IID construction and Dirichlet sweeps would strengthen the evaluation. It would also be beneficial to report statistical variance of experiments (mean and standard deviation).

- **Communication accounting transparency.** It would help to specify exactly what is counted in the communication budget (seed, indices/metadata, number of bits per scalar, any error-feedback state, whether downlink is counted, participation $(\beta)$).

---

### References
[1] Nagaraj et al. "TOFU: Toward Obfuscated Federated Updates by Encoding Weight Updates Into Gradients From Proxy Data." (2024)
[2] Zhou et al. "Distilled one-shot federated learning."  (2020).

**Questions:**

1. **Privacy (formal vs obfuscation).** Is the intended claim formal (e.g., DP / information-theoretic), or explicitly “obfuscation only”? If formal, what is the guarantee and adversary model, and how is multi-round accumulation addressed?

2. **Unconstrained accuracy.** What is each method’s best-achieved accuracy (no byte cap)? How large must $(m)$ be for FedMPDD to close the gap to FedSGD? A small $(m \to d)$ sanity curve would be informative.

3. **Baselines (TOFU, DOSFL).** Could apples-to-apples results and a brief discussion of trade-offs clarify whether FedMPDD is strictly better, comparable, or worse on certain axes?

4. **Non-IID robustness + stats.** Would Dirichlet-$\alpha$ sweeps with mean±std across seeds, and per-client dataset statistics, help make the heterogeneity more transparent?

5. **Fixed-budget protocol.** For clarity, does the budget count uplink only (sum over participating clients × rounds) or also downlink, and what per-round byte formula is used for each method (so the allowed rounds under a cap can be independently computed)?

---

### Official Review · Reviewer_Aaub · 2025-10-28

**Soundness:** 2
**Presentation:** 1
**Contribution:** 3
**Rating:** 4
**Confidence:** 3

**Summary:**

This paper proposes FedMPDD, a novel federated learning algorithm that enhances both communication efficiency and privacy protection by employing multi-projected directional derivatives for gradient encoding. Clients compress their high-dimensional gradients into low-dimensional scalar projections along multiple random directions, reducing uplink communication from O(d) to O(m) where m is significantly smaller than d, while the server reconstructs the global update using shared random seeds. The approach provides inherent privacy against gradient inversion attacks through its rank-deficient projection mechanism and achieves a convergence rate of O(1/√K) comparable to FedSGD, with the number of projections m offering a tunable privacy-communication-accuracy trade-off. Extensive experiments demonstrate FedMPDD's superior performance in reducing communication costs by up to 356 times while maintaining model accuracy and delivering strong privacy protection, outperforming existing methods in resource-constrained federated learning scenarios.

**Strengths:**

1. This paper has a good starting point and a strong theoretical basis.
2. Experimental results show that this method can greatly improve the efficiency of federated communication.

**Weaknesses:**

1. The authors used a lightweight network in the experiment. When extended to a large network, is the proposed method effective? For example, resnet18,34,50,
2. As far as I know, there are many efficient communication federated learning methods. Why only compare with FedSGD? This limits the verification of the effectiveness of the proposed method.
3. The presentation of this paper needs to be greatly improved. There are many strange typographical and writing errors.

**Questions:**

Please see the weaknesses.

---

### Official Review · Reviewer_Bagi · 2025-10-29

**Soundness:** 3
**Presentation:** 3
**Contribution:** 2
**Rating:** 4
**Confidence:** 4

**Summary:**

This paper introduces FedMPDD (Federated Learning via Multi-Projected Directional Derivatives), a novel algorithm for federated learning that simultaneously addresses the dual challenges of high communication overhead and privacy vulnerabilities from Gradient Inversion Attacks. The core mechanism involves clients encoding their high-dimensional gradients ($O(d)$) by computing directional derivatives along $m$ different random vectors. Clients then upload only the $m$ resulting scalar projections and a single random seed. The server, having received the seed, regenerates the exact same random vectors locally and reconstructs an unbiased gradient estimator. Extensive experiments show that FedMPDD achieves high model accuracy under strict communication budgets.

**Strengths:**

1.The core idea of transmitting only $m$ scalar projections plus a single random seed is a clever trick. It allows the server to perfectly reconstruct the projection vectors without the $\mathcal{O}(md)$ cost of uploading them, achieving a true $\mathcal{O}(m)$ uplink cost.

2.The privacy guarantee stems from the fundamental rank-deficiency of the $m \ll d$ projection, creating a large nullspace that makes gradient inversion impossible. This is a privacy-by-design approach, and the paper makes a compelling case that its constant, magnitude-independent privacy is superior to the fluctuating guarantees of LDP.

3.The paper provides a solid theoretical foundation. The convergence analysis clearly identifies and solves the dimension-dependency problem of a single projection, ultimately proving the standard $O(1/\sqrt{K})$ convergence rate for FedMPDD.

4.The experiments are thorough and the results are outstanding. FedMPDD is shown to achieve target accuracy with dramatically less communication (e.g., 1.32 GB vs.\ 471.96 GB for FedSGD on CIFAR-10). Crucially, it does this while maintaining strong privacy (SSIM $< 0.22$), whereas other compression-first methods (QSGD, Top-k, lp-proj) are shown to be vulnerable to GIAs (SSIM $> 0.74$).

**Weaknesses:**

1.Algorithm 2 explicitly shows the client first computing the full $d$-dimensional gradient $g_i(x_k)$ and then performing $m$ dot products. This adds an $\mathcal{O}(md)$ computational cost on top of the standard backpropagation. The paper does address this in Remark 1 and Appendix F, suggesting a JVP-based projected-forward approach that avoids computing $g_i$ explicitly. However, it is not clear if this optimization was used in the main experiments. If not, the $\mathcal{O}(md)$ cost (or the cost of $m$ JVP passes) could be a significant bottleneck for resource-constrained clients, especially as $m$ grows.

2.The parameter $m$ is the crucial knob for the paper's trade-offs. The paper states $m$ can be logarithmic in $d$ per the JL Lemma, but the experimental values (e.g., 400, 800, 2000) seem to be hand-tuned. Table A.9 shows that performance is highly sensitive to this, with accuracy collapsing from 75.02\% to 30.44\% when $m$ is reduced from 200 to 50. The paper would be stronger with a more principled discussion of how $m$ should be selected in practice for a new task.

3.Experimental scope is limited to small vision benchmarks and modest models; absence of results on ResNet-18/50 or transformers and real federated workloads leaves scalability and utility under realistic heterogeneity unclear.


4.Comparisons omit secure aggregation and strong DP-SGD baselines calibrated to similar accuracy loss, as well as advanced compression with error-feedback and modern sketching tuned for low distortion.

5.Privacy evaluation relies heavily on SSIM of reconstructions, which may not fully capture leakage; broader attack metrics such as label leakage, membership inference, and success rates under adaptive adversaries would increase confidence.

**Questions:**

1.The privacy bound $T < d/m$ seems to be the main theoretical limitation. Can you comment on the practical privacy guarantees when an adversary observes a client for $T$ rounds where $T \times m > d$?

2.The results seem highly sensitive to $m$, as shown in Table A.9. How were the $m$ values (e.g., $m = 600$, $m = 2000$) for the main experiments in Tables 1 and 2 chosen? Is there a heuristic or a more principled way to set this crucial hyperparameter?

3.Abstract Typo: please confirm that the $O(1/K)$ convergence rate mentioned in the abstract is a typo and that the correct proven rate is $O(1/\sqrt{K})$.

4.How does FedMPDD interact with secure aggregation, personalization layers, or error-feedback, and could these be combined to further reduce leakage while improving convergence on non-IID data ?

---

### Note · Authors · 2025-11-13

I have read and agree with the venue's withdrawal policy on behalf of myself and my co-authors.